# A MademoiseLLE domain binding platform links the key RNA transporter to endosomes

**Senthil-Kumar Devan**[1‡], **Stephan Schott-Verdugo**[2,3‡], **Kira Müntjes**[1], **Lilli Bismar**[1], **Jens Reiners**[4], **Eymen Hachani**[5], **Lutz Schmitt**[5], **Astrid Höppner**[4], **Sander HJ Smits**[4,5]*, **Holger Gohlke**[2,3]*, **Michael Feldbrügge**[1]*

**1** Institute of Microbiology, Heinrich Heine University Düsseldorf, Cluster of Excellence on Plant Sciences, Düsseldorf, Germany, **2** John von Neumann Institute for Computing (NIC), Jülich Supercomputing Centre (JSC), Institute of Biological Information Processing (IBI-7: Structural Bioinformatics), and Institute of Bio- and Geosciences (IBG-4: Bioinformatics), Forschungszentrum Jülich GmbH, Jülich, Germany, **3** Institute for Pharmaceutical and Medicinal Chemistry, Heinrich Heine University Düsseldorf, Düsseldorf, Germany, **4** Center for Structural Studies, Heinrich Heine University Düsseldorf, Düsseldorf, Germany, **5** Institute of Biochemistry I, Heinrich Heine University Düsseldorf, Düsseldorf, Germany

‡ These authors share first authorship on this work.
* sander.smits@hhu.de (SHJS); gohlke@uni-duesseldorf.de, h.gohlke@fz-juelich.de (HG); feldbrue@hhu.de (MF)

## Abstract

Spatiotemporal expression can be achieved by transport and translation of mRNAs at defined subcellular sites. An emerging mechanism mediating mRNA trafficking is microtubule-dependent co-transport on shuttling endosomes. Although progress has been made in identifying various components of the endosomal mRNA transport machinery, a mechanistic understanding of how these RNA-binding proteins are connected to endosomes is still lacking. Here, we demonstrate that a flexible MademoiseLLE (MLLE) domain platform within RNA-binding protein Rrm4 of *Ustilago maydis* is crucial for endosomal attachment. Our structure/function analysis uncovered three MLLE domains at the C-terminus of Rrm4 with a functionally defined hierarchy. MLLE3 recognises two PAM2-like sequences of the adaptor protein Upa1 and is essential for endosomal shuttling of Rrm4. MLLE1 and MLLE2 are most likely accessory domains exhibiting a variable binding mode for interaction with currently unknown partners. Thus, endosomal attachment of the mRNA transporter is orchestrated by a sophisticated MLLE domain binding platform.

## Author summary

Eukaryotic cells rely on sophisticated intracellular logistics. Macromolecules like mRNA must be transported to defined subcellular destinations for local translation. This is mediated by active transport along the cytoskeleton. Endosomes are carrier vehicles that shuttle along microtubules by the action of molecular motors. It is currently unclear how mRNAs are attached mechanistically to these membranous units during transport. We study the model microorganism *Ustilago maydis* where numerous components of endosomal mRNA transport have already been identified. Previously, we found that the key RNA-binding protein Rrm4 interacts with the endosomal adaptor protein Upa1. Here, we

**Data Availability Statement:** The authors confirm that all data underlying the findings are fully available without restriction. All relevant data are within the paper and its Supporting Information

files. The MLLE2 structure was deposited at the worldwide protein data bank under the accession code 7PZE. We uploaded the data to the Small Angle Scattering Biological Data Bank (SASBDB) with the accession codes SASDMS5(G-Rrm4) and SASDMT5 (H-Rrm4-NT4). https://www.sasbdb.org/data/SASDMS5/ https://www.sasbdb.org/data/SASDMT5/.

**Funding:** The work was funded by grants from the Deutsche Forschungsgemeinschaft under Germany's Excellence Strategy EXC-2048/1 - Project ID 39068111 to MF; Project-ID 267205415 – SFB 1208 to MF (project A09), HG (project A03), and LS (project A01). The Center for Structural Studies was funded by the Deutsche Forschungsgemeinschaft (DFG Grant number 417919780; INST 208/740-1 FUGG; INST 208/761-1 FUGG to S.H.J S). The funders had no role in study design, data collection and analysis, decision to publish, or preparation of the manuscript.

**Competing interests:** The authors have declared that no competing interests exist.

perform a structure/function analysis and discovered that Rrm4 contains not one but three different versions of a protein-protein interaction domain, called the MademoiseLLE domain, to facilitate the attachment with transport endosomes. Importantly, they function with a strict hierarchy with one essential domain and the others play accessory roles. This is currently, the most detailed mechanistic description of how an RNA-binding protein and its bound cargo mRNAs are attached to endosomes. The usage of three similar protein-protein interaction domains forming a complex binding platform with a defined hierarchy might be operational also in other unknown protein-protein interactions.

## Introduction

mRNA localisation and local translation are essential for spatiotemporal control of protein expression. An important mechanism to achieve localised translation is the active transport of mRNAs along the cytoskeleton [1–3]. Mainly, long-distance transport of mRNA is mediated by motor-dependent movement along microtubules. Transport endosomes are important carriers that move messenger ribonucleoprotein complexes (mRNPs), consisting of RNA-binding proteins and cargo mRNAs on their cytoplasmic surface [1,4,5]. This process is evolutionarily conserved in fungi, plants, and animals [5–11]. In endosperm cells of developing rice seeds, cargo mRNAs are transported to the cortical endoplasmic reticulum (ER) by the action of the two RNA recognition motif (RRM)-containing proteins RBP-P and RBP-L. These form a quaternary complex with membrane trafficking factor NSF (*N*-ethylmaleimide-sensitive factor) and small GTPase Rab5a on the endosomal surface [12]. In neurons, mRNA transport has been linked to early and late endosomes as well as lysosomal vesicles. Especially, local translation of mRNAs encoding mitochondrial proteins on the surface of late endosomes is needed for mitochondrial function. Importantly, this trafficking process has been associated with the neuronal Charcot Marie-Tooth disease [8]. Annexin 11, a factor implicated in amyotrophic lateral sclerosis (ALS), was found as an mRNP linker on motile lysosomal vesicles [9]. Also, the five-membered FERRY complex was recently identified connecting mRNAs encoding mitochondrial proteins to neuronal endosomes by interaction with the active form of Rab5 [10,11].

Among the best-studied examples of membrane-coupled mRNA transport is the endosomal mRNA transport in the corn pathogen *Ustilago maydis* [5,13,14]. Extensive peripheral movement of mRNAs is needed for efficient unipolar growth of infectious hyphae. These hyphae grow highly polarised by expanding at the growing tip and inserting regularly spaced septa at the basal pole. Loss of mRNA distribution causes aberrant bipolar growth [6,15,16]. Key vehicles of cargo mRNAs are Rab5a-positive endosomes that shuttle along microtubules by the concerted action of plus-end directed kinesin-3 and minus-end directed cytoplasmic dynein [6]. Important cargo mRNAs are, for example, all four septin mRNAs. Their local translation during transport is essential to form heteromeric septin complexes on the surface of transport endosomes. Endosomes deliver these complexes to the hyphal tip, forming a defined gradient of septin filaments at the growing pole [17–19].

Rrm4 is the key RNA-binding protein of the transport process that recognises defined sets of cargo mRNAs via its three N-terminal RRMs (Fig 1A) [19]. Rrm4 and bound cargo mRNAs are linked to endosomes by Upa1, containing a FYVE zinc finger for interaction with PI$_3$P lipids (phosphatidylinositol 3-phosphate; Fig 1A) [16,20]. The adaptor protein Upa1 contains a PAM2 motif (poly[A]-binding protein interacting motif 2) [21–23] and two PAM2-like (PAM2L) sequences. These motifs are crucial for interaction with MademoiseLLE (MLLE) domains of the poly(A)-binding protein Pab1 and Rrm4, respectively (Fig 1A) [16].

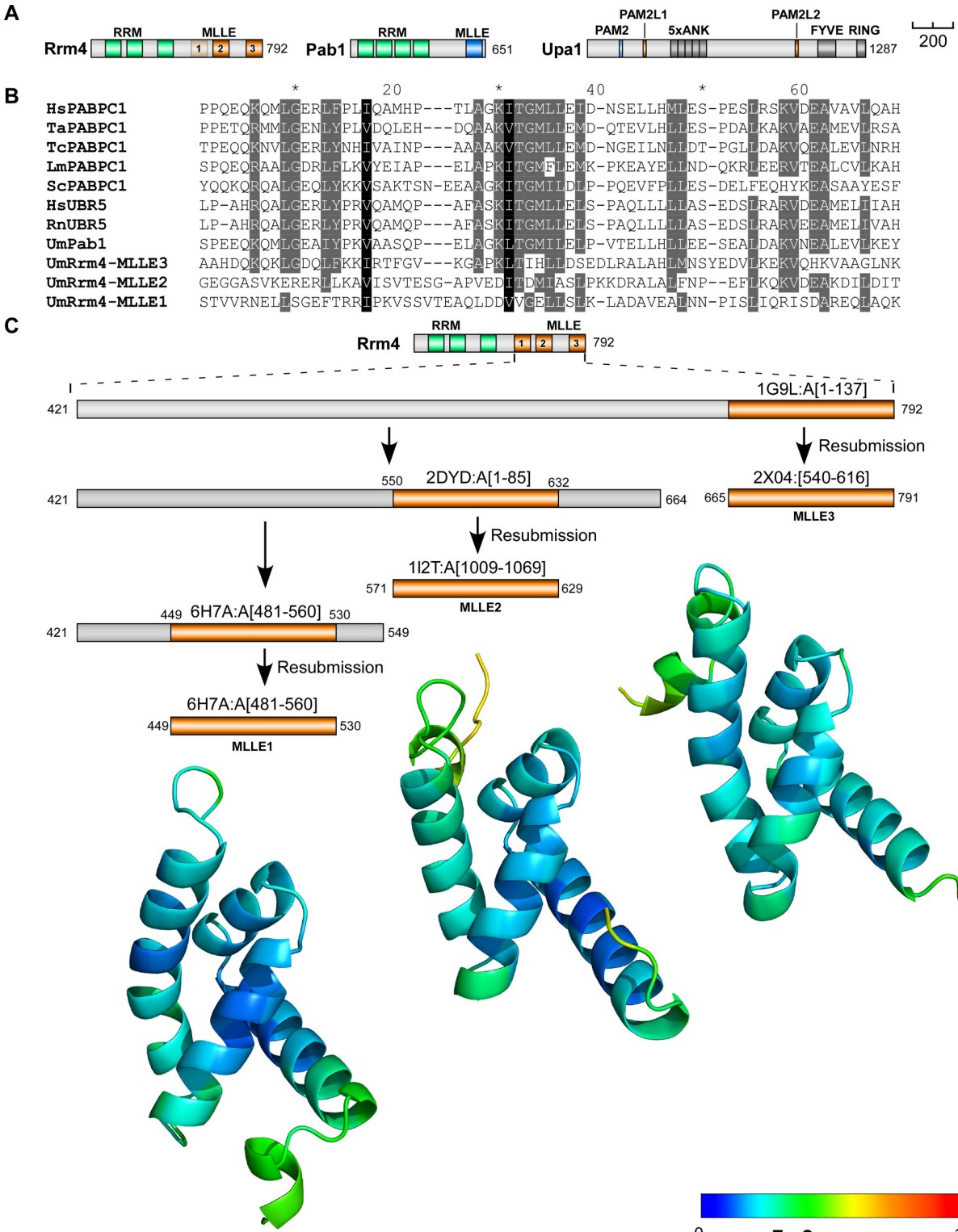

**Fig 1. The C-terminal half of Rrm4 contains three MLLE domains.** (**A**) Schematic representation of protein variants drawn to scale (bar, 200 amino acids, number of amino acids indicated next to protein bars) using the following colouring: dark green, RNA recognition motif (RRM); orange, MLLE^Rrm4 domains; dark blue, MLLE^Pab1; light blue PAM2; light orange PAM2L sequence (PL1–2) Ankyrin repeats (5xANK), FYVE domain, and RING domain of Upa1 are given in dark grey. (**B**) Sequence alignment of previously determined MLLE domains showing the degree of similarity to the three Rrm4-MLLE domains and the positions (Hs—*Homo sapiens*, Ta—*Triticum aestivum*, La—*Leishmania major*, Sc—*Saccharomyces cerevisiae*, Tc—*Trypanosoma cruzi*, Rn—*Rattus norvegicus*, Um—*Ustilago maydis*, PABPC1, Pab1 –poly [A]-binding protein, UBR5—E3 ubiquitin-protein ligase). Accession number and sequence coverage are listed in S1 Table.

Multiple sequence alignment was performed by ClustalW. (**C**) Identification and modelling of C-terminal MLLE domains of Rrm4. The iterative process is depicted graphically. The best-identified template for each run, and the region of that template that aligns with Rrm4, are displayed (see also S1A Fig for the templates used for the final models). The structural models obtained are shown for the span of the first identified template and are coloured according to their per-residue TopScore, where the scale from 0 to 1 indicates a low to high local structural error.

The MLLE domain was first identified as a conserved domain at the C-terminus of the human cytoplasmic poly(A)-binding protein 1 (PABP1C) [24,25]. Solution and crystal structures of PABC domains from PABP1C and ubiquitin ligase UBR5 showed that they are structurally conserved [26,27]. The domain is about 70 amino acids in length and consists of five bundled α-helices. Interaction with the PAM2-binding motif (consensus sequence xxLNxxAxEFxP) is characterised by the central α-helix 3 with the sequence KITGMLLE and mediated by two adjacent hydrophobic pockets [28], with the binding of the Phe residue of the PAM2 motif being the major determinant for this interaction [29]. Besides human PABPC1, there are currently only two additional proteins with MLLE domains described: the ubiquitin ligase UBR5 functioning, for example, during microRNA-mediated gene silencing [30] and Rrm4-type RNA-binding proteins from fungi (Fig 1B) [31].

Mutations in the C-terminal MLLE domain of Rrm4 result in the loss of Rrm4 motility, suggesting that the link to endosomes is disrupted [15]. Consistently, the C-terminus of Rrm4 recognises the PAM2L sequence of the adaptor protein Upa1 [16], suggesting that the interaction of MLLE domains with PAM2L sequences is responsible for its endosome association. This study combines structural biology with fungal genetics to demonstrate that the C-terminal half of Rrm4 has three divergent MLLE domains with a flexible arrangement and each domain contributes differentially to the endosomal attachment.

## Results

### Iterative structural modelling predicts three MLLE domains at the C-terminus of Rrm4

To generate structural models of the MLLE domains present in Rrm4, we focused on the C-terminal part of the protein (residues 421 to 792). This excluded the three N-terminal RRMs but included the previously predicted two C-terminal MLLE domains (Fig 1A and 1B) [31]. Subjecting this Rrm4 sequence region to iterative comparative modelling with TopModel (Fig 1C) [32] revealed, as expected, the previously identified two regions with homology for MLLE domains located at residues 571–629 and 712–791 (denoted MLLE2 and MLLE3; Fig 1C) [31]. Unexpectedly, using the TopModel workflow with its efficient template selection capabilities [32], we identified an additional *de novo* predicted MLLE domain located at residues 451–529 (denoted MLLE1; Figs 1B, 1C and S1A). Although the sequence identity between templates and their respective Rrm4 sequence stretches was only 17 to 32% (Figs 1B and S1A), the generated MLLE domain models had a high predicted local structural quality, as assessed by TopScore (Fig 1C) [33]. The generated models were also verified by the current deep neural network modelling approaches AlphaFold2 and RoseTTAFold (S1B Fig) [34,35], further indicating that the C-terminal half of Rrm4 has three MLLE domains instead of the previously identified two. All of these MLLE domains might be relevant for the interaction with Upa1.

### X-ray analysis of the second MLLE domain confirms the predicted structural models

To verify the structural models further, we expressed and purified an N-terminally truncated version of the Rrm4 carrying the three MLLE domains in *Escherichia coli* (S2A and S2B Fig;

version H-Rrm4-NT4 carrying an N-terminal hexa-histidine-tag; Materials and methods) [16]. Size exclusion chromatography combined with Multi-angle light scattering (MALS) indicated that the protein was homogenous and did not form aggregates (S2C Fig). We thus set out to crystallize the protein for X-ray diffraction analysis (see Material and methods). Testing 2016 different conditions, crystals were only obtained in individual cases after at least 7 days of incubation. A complete dataset was collected from a single crystal diffracting to 2.6 Å resolution and a $P4_32_12$ symmetry. Data and refinement statistics are given in S2 Table. Surprisingly, the unit cell dimensions were small and, with a Matthews coefficient assuming 50% solvent content, only 128 amino acids would fit into the asymmetric unit of the crystal. Hence, the unit cell had an insufficient size to cover H-Rrm4-NT4, which contains 380 amino acids. Using the predicted models of MLLE1-3 as templates for molecular replacement, only MLLE2 gave a clear solution, showing after refinement that two copies of MLLE2 (residues 567–630) were present in the asymmetric unit. For comparison, previously, two copies of the MLLE domain in the asymmetric unit were reported in crystals of MLLE of UBR5 [36]. The structural data indicated that the protein was truncated from both termini during crystallisation, resulting in a shortened version of the H-Rrm4-NT4 protein that formed stable crystals (see Material and methods). Both MLLE2$^{Rrm4}$ copies adopted the same overall fold as seen by the RMSD of 0.29 Å over 59 C-alpha atoms. The MLLE2$^{Rrm4}$ crystal structure displayed high similarity with the MLLE domain of the ubiquitin ligase UBR5 (MLLE$^{UBR5}$; PDB code 3NTW, RMSD of 0.97 Å over 56 amino acids) [36] and the MLLE domain of PABPC1 (PDB code 3KUS, RMSD of 1.34 Å over 61 amino acids) [37]. The MLLE2$^{Rrm4}$ domain consisted of four helices (designated α2–5; Fig 2A) arranged as a right-handed superhelix similar to MLLE$^{UBR5}$. In comparison to the MLLE domain of PABPC1, the first short helix was absent in both MLLE2$^{Rrm4}$ and MLLE$^{UBR5}$ structures.

When comparing the obtained crystal structure with the MLLE2$^{Rrm4}$ model generated by TopModel, the average RMSD was 0.69 Å over the backbone atoms, close to the uncertainty of the atomic coordinates of the experimental structure (Fig 2B). Importantly, this confirmed our structural model of MLLE2$^{Rrm4}$ and strongly suggested that the modelled MLLE1$^{Rrm4}$ and MLLE3$^{Rrm4}$ domains should be of equally high quality.

We compared the predicted models of MLLE1-3$^{Rrm4}$ with the known structure of the human PABPC1 focusing on the well-described PAM2 peptide-binding pocket. This revealed that MLLE3$^{Rrm4}$ maintained a characteristic Gly residue at position 736 that binds the conserved Phe residue of the PAM2 motifs, a major binding determinant in PABPC1 and UBR5 (Fig 2C) [29,36]. However, the binding interfaces of MLLE1$^{Rrm4}$ and MLLE2$^{Rrm4}$ were altered compared to the 'canonical' binding site in PABPC1 and UBR5 (Fig 2C). Instead of Gly, MLLE1$^{Rrm4}$ and MLLE2$^{Rrm4}$ had a Ser and Arg in the corresponding positions 471 and 573. The notion that MLLE1$^{Rrm4}$ and MLLE2$^{Rrm4}$ may differ from canonical MLLE domains was also supported by the lower sequence identity of MLLE1$^{Rrm4}$ and MLLE2$^{Rrm4}$ when compared to previously characterised MLLE domains (Figs 1B; S1A). In summary, structural modelling revealed the presence of three MLLE domains at the C-terminus of Rrm4. Furthermore, the structure of the MLLE2$^{Rrm4}$ domain was successfully verified by X-ray crystallographic analysis. MLLE1$^{Rrm4}$ and MLLE2$^{Rrm4}$ are divergent in the key region of PAM2 binding, suggesting that these domains might employ a different binding mode or show a different binding specificity.

## The MLLE domains of Rrm4 form a binding platform with flexible arrangement of the individual domains

To study the relative arrangement of all three MLLE$^{Rrm4}$ domains and the orientation to the N-terminal RRMs, we performed Small-Angle X-ray Scattering (SAXS) experiments. We

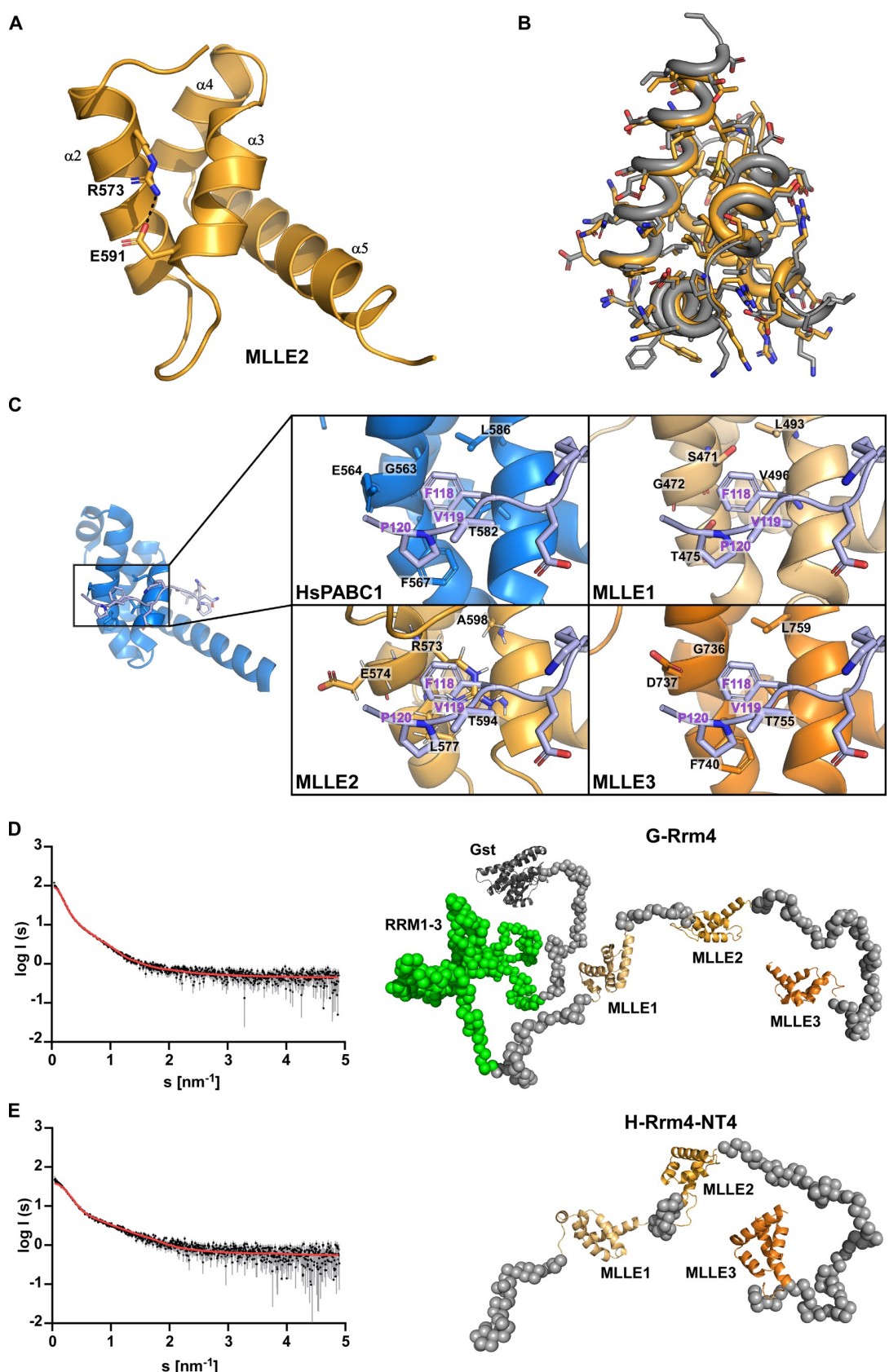

**Fig 2. Rrm4 contains a C-terminal tripartite MLLE binding platform.** (**A**) Crystal structure of the MLLE2 domain is highlighted in orange. The four helices are indicated by α2–5 according to the 5 helix nomenclature found in MLLE domains (28). Note that the first short helix α1 is missing. Arg573 and Glu591 are highlighted in the sticks. These sides chains would interfere with the binding of the canonical Phe of PAM2 type motifs. (**B**) Structural alignment of the MLLE2 model generated by TopModel and the X-ray crystal structure of this domain (grey or orange, respectively). The all-atom RMSD is 0.69 Å, resulting mostly from different rotamers of solvent-exposed sidechains. (**C**) Comparison of peptide-binding sites after structural alignment of the models of Rrm4 MLLE domains (orange shades) and the canonical MLLE domain of HsPABPC1 (blue; PDB ID 3KUS) and manually placing the PAM2 motif of PAIP2 (lilac). In the interaction of MLLE$^{PABPC1}$ with PAM2 of PAIP2, Phe118 of PAM2 is the major determinant for binding and present in all the PAM2 motifs except LARP4a and b ([28,29]; S3A Fig). Of the identified Rrm4 MLLE domains, only MLLE3 retains all sidechains that favour the binding of this characteristic Phe; particularly, Gly736 should allow the Phe to bind into a pocket. MLLE1 and MLLE2 have Ser471 and Arg573 instead of Gly in this position, suggesting that Phe binding would be sterically hindered in these interfaces. (**D**) *Left panel* Experimental data curve for GST-Rrm4 is shown in black dots with grey error bars, the EOM fit as a red line ($\chi^2$ = 1.289). The intensity is displayed as a function of momentum transfers. *Right panel* Selected model of the EOM analysis from GST-Rrm4 with a $R_g$ of 8.75 nm, a $D_{max}$ of 23.99 nm with a volume fraction of ~0.25. (**E**) *left panel* Experimental data curve for H-Rrm4NT4 is shown in black dots with grey error bars, the EOM fit as the red line ($\chi^2$ = 1.262). The intensity is displayed as a function of momentum transfers. *right panel* Selected model of the EOM analysis from H-Rrm4NT4 with a $R_g$ of 5.10 nm, a $D_{max}$ of 16.43 nm, and a volume fraction of ~0.75. The MLLE subdomains are shown in cartoon representation (MLLE1 in light orange, MLLE2 in orange, MLLE 3 in dark orange, and the GST in dark grey) and the missing amino acids as grey spheres (all other models and the SAXS data are available in S2E Fig).

expressed and purified H-Rrm4-NT4 as well as the full-length protein with N-terminal GST fusion (glutathione S-transferase; G-Rrm4) from *E. coli* (see Materials and methods). Primary data analysis of the scattering curves [38,39] revealed that both proteins were monomeric and highly flexible in solution (S3 Table; Figs 2D–2E; S2D–S2E). To visualise the different protein conformations, we performed an Ensemble Optimization Method (EOM) analysis for both the G-Rrm4 and H-Rrm4-NT4 proteins (Fig 2D–2E). We used our MLLE models and a GST model (PDB entry: 1ua5) together with the protein sequence for G-Rrm4 as input, yielding a distribution of different conformations of the protein in solution (representative models in S2E Fig). One model of G-Rrm4, representing 25% of the population, revealed that the C-terminal part containing MLLE1-3$^{Rrm4}$ adopted an elongated and mainly unfolded but open conformation (Figs 2D; S2D–S2E). The N-terminal part, containing RRM domains of the GST fusion protein, adopted a more globular structure, indicating less flexibility within this region (Figs 2D; S2D–S2E). Studying only the C-terminal part of Rrm4 revealed that the most prominent model of this analysis (75% of the population) had a nearly identical conformation as the one selected for G-Rrm4 (Figs 2D–2E, S2E). This suggests that the C- terminal part of Rrm4-NT4 adopts a very similar orientation when expressed by itself. This analysis deduced that the MLLE1-3$^{Rrm4}$ domains form a C-terminal binding platform with a flexible arrangement for multiple contact sites for binding partners. Thus, the RRM domains for RNA interaction are spatially separated from the protein interaction platform.

## The third MLLE is essential for interaction with PAM2-like sequences of Upa1

To evaluate the interaction capacity of MLLE1-3$^{Rrm4}$, we performed *in vitro* binding studies. We expressed different deletion versions of Rrm4 as N-terminal GST fusions in *E. coli*. As a control, we expressed an N-terminal GST fusion of the MLLE domain of Pab1 (Fig 3A; Materials and methods). To check the physical interaction with PAM2 and PAM2L sequences of Upa1, we expressed 18 amino acid fragments (Fig 3A) as N-terminal hexa-histidine-SUMO (HS) fusion proteins (see Materials and methods). In GST pull-down experiments using GST fusion proteins as bait, G-Pab1-MLLE interacted with HS-PAM2 but not with the HS-PAM2L motifs of Upa1 (S3B Fig, lane 2). Conversely, G-Rrm4-NT4 recognised the two HS-PAM2L motifs of Upa1 but not the HS-PAM2 motif (S3B Fig, lane 3) [16]. Interestingly, the interaction

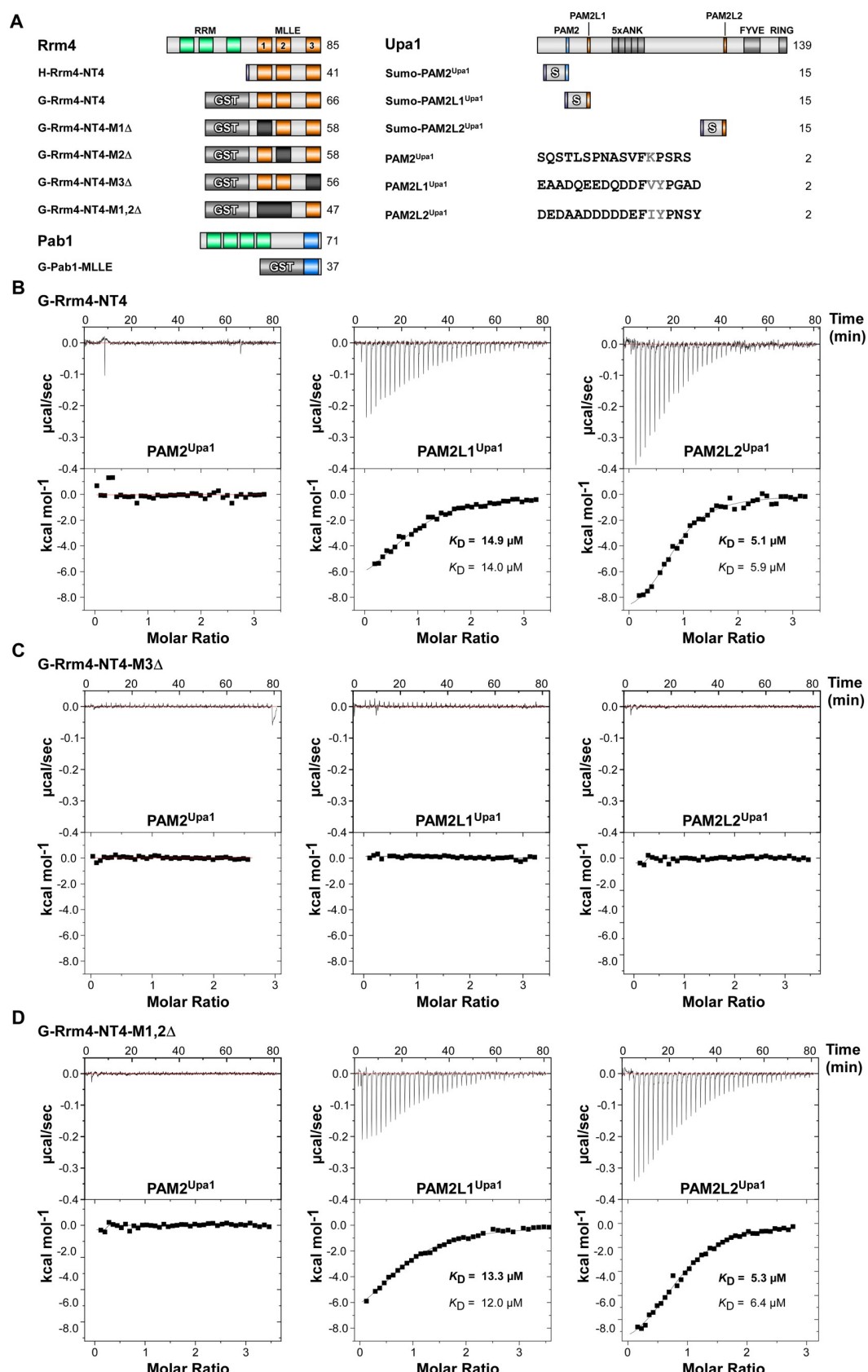

**Fig 3. MLLE3$^{Rrm4}$ is crucial for PAM2L1$^{Upa1}$ and PAM2L2$^{Upa1}$ binding.** (**A**) Schematic representation of protein variants (Molecular weight in kilo Dalton indicated) using the following colouring: dark green, RNA recognition motif (RRM); orange, MLLE$^{Rrm4}$ domains; dark blue, MLLE$^{Pab1}$; light blue PAM2; light orange PAM2L sequence (PL1–2). Ankyrin repeats (5xANK), FYVE domain, and RING domain of Upa1 are given in dark grey. GST and SUMO tags are labelled. Variant amino acids of the FxP and FxxP of PAM2 and PAM2L sequences are printed in grey font. (**B-D**) Representative isothermal titration calorimetry (ITC) binding curves of MLLE domains. Experiments were performed using GST- or Histidine-tagged MLLE variants and synthetic PAM2 peptide variants. $K_D$ values of two independent measurements are given (values corresponding to the indicated data are given in bold).

with both PAM2L motifs was lost when MLLE3$^{Rrm4}$ was deleted (G-Rrm4-NT4-M3Δ; S3B Fig, lane 6), while constructs with deletion of MLLE1$^{Rrm4}$ or MLLE2$^{Rrm4}$, or both MLLE1$^{Rrm4}$ and MLLE2$^{Rrm4}$, still interacted with the HS-PAM2L motifs of Upa1 (S3B Fig, lane 4,5 and 7).

To validate qualitatively whether these results also hold true for full-length proteins, we performed yeast two-hybrid experiments comparable to previous studies [16]. To this end, Upa1 or Rrm4 versions were fused at the N-terminus with the DNA-binding domain (BD) and activation domain (AD) of Gal4p, respectively (see Materials and methods; the C-termini were fused with the enhanced version of the green fluorescent protein [Gfp], Clontech; or the monomeric version of red fluorescent protein mKate2 [Kat], respectively) [16,40]. Rrm4-Kat interacted with full-length Upa1-Gfp (S4A Fig) [16] and interaction was lost when MLLE3$^{Rrm4}$ was deleted. Mutations in MLLE1$^{Rrm4}$, MLLE2$^{Rrm4}$ or MLLE1,2$^{Rrm4}$ did not alter the interaction with Upa1-Gfp (S4B–S4D Fig). To further investigate the presence of unknown interaction motifs in Upa1-Gfp, variants carrying block mutations in either or both PAM2L1 and PAM2L2 motifs were tested against the Rrm4-Kat versions (S4B and S4C Fig). Upa1-Gfp versions with block mutations in either PAM2L1 or PAM2L2 still interacted with the Rrm4-Kat versions (S4D Fig). However, when both PAM2L1,2 motifs were mutated, the interaction between the Upa1 and Rrm4 was lost, comparable to earlier observation (S4D Fig) [16]. Invariably, MLLE3$^{Rrm4}$ deletion caused loss of interaction with all Upa1-Gfp versions (S4A–S4D Fig). These results confirm that the MLLE3$^{Rrm4}$ domain is essential for the interaction with Upa1. MLLE1$^{Rrm4}$ and MLLE2$^{Rrm4}$ appear to be dispensable for the interaction with Upa1 suggesting the presence of additional interaction partners (see below).

To obtain quantitative data on the protein/peptide interactions, we performed isothermal titration calorimetry (ITC) experiments with purified proteins (S6A Fig) and synthetic peptides with a length of 18 amino acids (PAM2$^{Upa1}$, PAM2L1$^{Upa1}$, and PAM2L2$^{Upa1}$; Fig 3A). The binding constant $K_D$ and the binding stoichiometry were calculated from the curves, which in all cases indicated a 1:1 ratio between the G-Rrm4-NT4 protein and the binding partner.

Testing G-Pab1-MLLE with the peptides revealed a $K_D$ of 14.6 μM for PAM2$^{Upa1}$ (S5A and S5B Fig), which is within the range of observed $K_D$ of 0.2 to 40 μM for known MLLE/PAM2 interactions like the MLLE domain of PABPC1 with various PAM2 sequences [41]. Testing G-Pab1-MLLE with PAM2L1$^{Upa1}$ and PAM2L2$^{Upa1}$ peptides, no indication for binding was observed. PAM2L sequences are rich in acidic residues and exhibit a different FxxP spacing than the canonical FxP sequence of PAM2 sequences in the core region (Figs 3A, S5A; see Discussion). The observed binding behaviour indicated a clear binding specificity differentiating PAM2 and PAM2L peptides. This was in line with the previously published GST pull-down experiments [16].

In comparison, testing G-Rrm4-NT4 with the peptides revealed a $K_D$ of 14.9 μM for PAM2L1$^{Upa1}$ and 5.1 μM for PAM2L2$^{Upa1}$ and no binding to PAM2$^{Upa1}$ (Fig 3B). This suggested a similar affinity when compared to the interactions of MLLE$^{Pab1}$ with PAM2 and demonstrated the high sequence specificity of the MLLE domains to their respective PAM2L sequences (see Discussion).

Analysing G-Rrm4-NT4-M3Δ with a deletion of MLLE3$^{Rrm4}$ revealed that binding to PAM2L1$^{Upa1}$ and PAM2L2$^{Upa1}$ was no longer detectable (Fig 3C). This was in line with

observations from the GST pull-down experiments (S5B and S5C Fig). This suggests that MLLE3$^{Rrm4}$ is essential for binding. Testing G-Rrm4-NT4 versions carrying deletions in either MLLE1$^{Rrm4}$ or MLLE2$^{Rrm4}$ showed no difference in binding affinity (S3F and S3G Fig). Even testing G-Rrm4-NT4 with a deletion in both MLLE1,2 domains exhibited a binding affinity in the same range as the wild type version containing all three MLLEs (Fig 3D). We conclude that (i) MLLE3$^{Rrm4}$ is vital for recognising PAM2L sequences with a higher affinity to PAM2L2 and (ii) neither MLLE1$^{Rrm4}$ nor MLLE2$^{Rrm4}$ contributed to the binding of PAM2L or PAM2 motifs (Figs 3, S3–S5; summarised in S6B Fig, see Materials and methods). This is consistent with our structural analysis revealing the differences in the binding site for these MLLE domains (see Discussion).

## The third MLLE domain of Rrm4 is essential for its function

To address how the different MLLE domains contribute to the biological function of Rrm4, we generated *U. maydis* strains carrying deletions in the respective domains of Rrm4 (Fig 4A). As genetic background, we used laboratory strain AB33, expressing the heteromeric master transcription factor of hyphal growth (bE/bW) under control of the nitrate inducible promoter P$_{nar1}$. Thereby, polar hyphal growth can be elicited efficiently and in a highly reproducible fashion by changing the nitrogen source (Fig 4B, top) [42]. To investigate dynamic endosomal transport, we used strains expressing functional C-terminal fusion Upa1-Gfp and Rrm4-Kat (see Materials and methods).

The resulting hyphae grew with a defined axis of polarity, i.e., they expanded at the hyphal tip and inserted basal septa leading to the formation of empty sections (Fig 4B and 4C). Loss of Rrm4 (*rrm4Δ* strain) caused the formation of hyphae growing at both ends, characteristic of aberrant bipolar growth (Fig 4B and 4C) [16]. Rrm4-Kat versions carrying deletions of MLLE1$^{Rrm4}$ or MLLE2$^{Rrm4}$ did not cause alterations in unipolar growth (Fig 4B and 4C). Furthermore, endosomal shuttling and co-localisation were indistinguishable from wild type (Fig 4D and 4E). Also, the number of endosomes (number of signals / 10 μm, S7A Fig), velocity, and processivity (S7B and S7C Fig) were comparable to wild type. Hence, the first two MLLE domains were dispensable for polar growth and endosomal shuttling under optimal growth conditions. Since the deletion of the first two MLLEs did not substantially alter the function of Rrm4, we infer that the deletion neither affected the overall structure of the protein nor interfered with other domains like the RNA-binding domain of the protein. This supports the conclusions of our biochemical experiments (see above).

Importantly, testing strains expressing Rrm4-Kat with deletion of the third MLLE domain revealed a loss-of-function phenotype similar to *rrm4Δ* strains. The number of bipolar hyphae was comparable to *rrm4Δ* strains (Fig 4B and 4C; mutation identical to allele *rrm4G$^{PA}$*) [15]. We observed drastic alteration in shuttling, and Rrm4 aggregates did not co-localise with motile Upa1-positive signals (Fig 4D and 4E). While the Rrm4 signals were static (Figs 4D and 4E; S4A–S4C), the number of motile Upa1-Gfp positive endosomes, their velocity, and their processivity were not affected (S7A–S7C Fig, summarised in S7D Fig). This is consistent with previous results showing that the third MLLE domain is important for the movement of Rrm4 and that endosomal shuttling of Upa1 is not affected if Rrm4 is missing [6,15,16]. To conclude, MLLE3$^{Rrm4}$ is an essential domain for Rrm4 attachment to endosomes in contrast to MLLE1$^{Rrm4}$ and MLLE2$^{Rrm4}$.

## The second MLLE domain plays accessory roles in endosomal Rrm4 attachment

To investigate the biological role of MLLE1$^{Rrm4}$ and MLLE2$^{Rrm4}$ in more detail, we generated strains expressing Rrm4-M1,2Δ-Kat, lacking both MLLE1$^{Rrm4}$ and MLLE2$^{Rrm4}$ domains, and

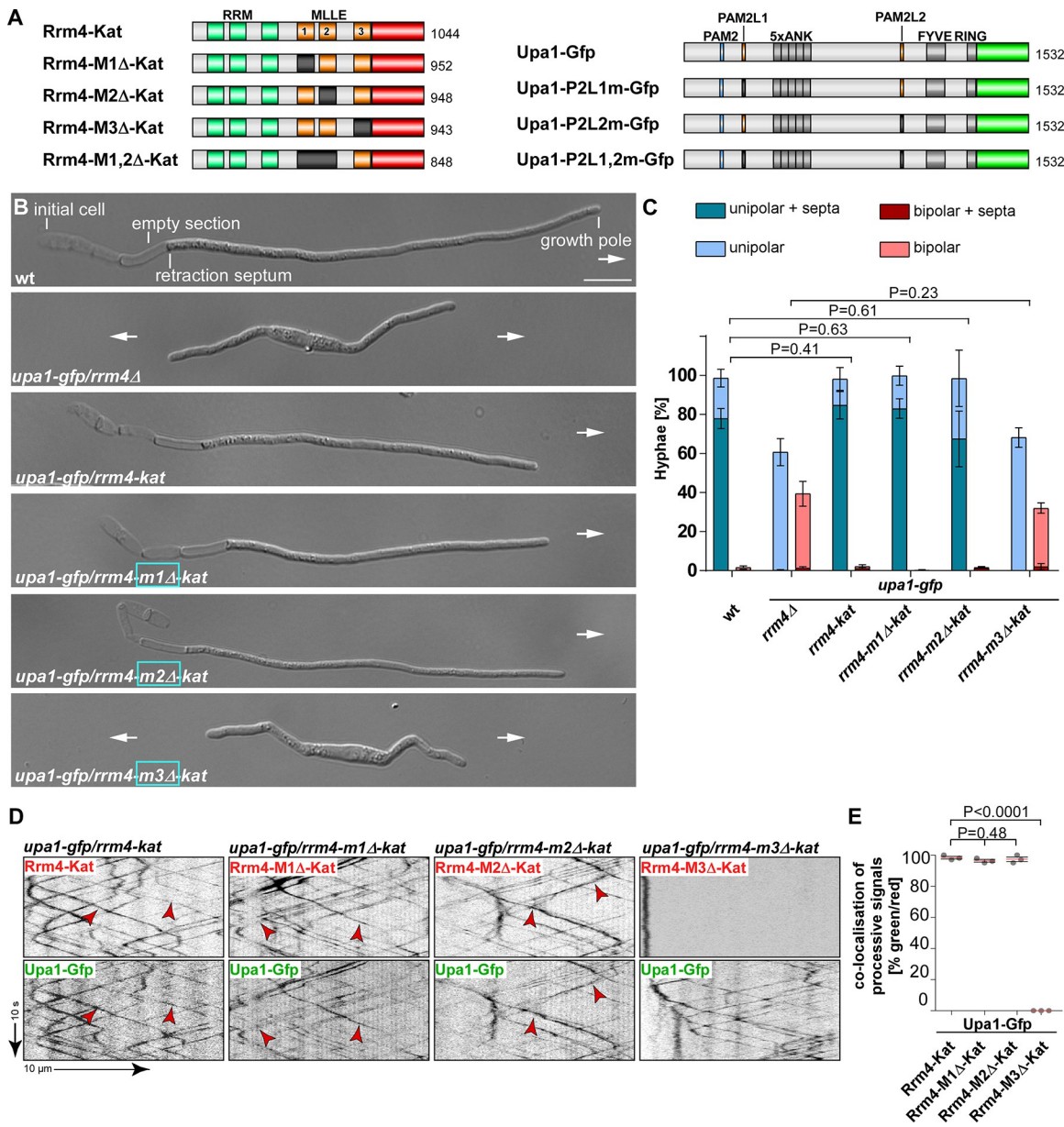

**Fig 4. MLLE3 is key for endosomal mRNA transport.** (**A**) Schematic representation of Rrm4 and Upa1 variants drawn not to scale (number of amino acids indicated next to protein bars) using the following colouring: dark green, RNA recognition motif (RRM); orange, MLLE domains; red, mKate2, blue, PAM2, light orange PAM2 like sequence (PL1–2) and light green, Gfp. Ankyrin repeats (5xANK), FYVE domain and RING domain of Upa1 are given in dark grey. (**B**) Growth of AB33 derivatives in their hyphal form (6 h.p. i.; size bar 10 μm). Growth direction is marked by arrows. (**C**) Quantification of hyphal growth of AB33 derivatives shown in B (6 h.p.i.): unipolarity, bipolarity and basal septum formation were quantified (error bars, SEM.; n = 3 independent experiments, > 100 hyphae were counted per strain; for statistical evaluation, the percentage of uni- and bipolarity was investigated by using unpaired two-tailed Student's t-test (α<0.05). (**D**) Kymographs of AB33 hyphae derivatives (6 h.p.i.; inverted fluorescence images) expressing pairs of red and green fluorescent proteins as indicated. Fluorescence signals were detected simultaneously using dual-view technology (arrow length on the left and bottom indicates time and distance, respectively). Processive co-localising signals are marked by red arrowheads. (**E**) Percentage of processive signals exhibiting co-localisation for strains shown in D (data points represent means from n = 3 independent experiments, with mean of means, red line and SEM; unpaired two-tailed Student's t-test (α<0.05); for each experiment, 10 hyphae per strains were analysed).

tested the influence on hyphal growth. Unipolar growth was not disturbed (Fig 5A and 5B). To challenge the endosomal attachment of Rrm4, we expressed Upa1-Gfp versions carrying mutations in PAM2L motif 1 or 2 as well as in both motifs; these motifs are important for Rrm4 interaction (Fig 4A) [16]. Strains expressing Rrm4-M1,2Δ-Kat in combination with mutated PAM2L1 or PAM2L2 of Upa1 showed unipolar growth comparable to wild type (Figs 5A and 5B; S8A and S8B), indicating that MLLE1Rrm4 and MLLE2Rrm4 were dispensable for unipolar growth even when the endosomal attachment was weakened by expressing Upa1 versions with mutated PAM2L motifs (S7D Fig). When studying Upa1 mutated in both PAM2L motifs, we observed an aberrant bipolar growth phenotype comparable to the upa1Δ strain (Fig 5A and 5B). This was expected, since the interaction of Rrm4 to endosomes is mediated by both PAM2L motifs [16]. Analysing Rrm4-M1,2Δ-Kat in this genetic background revealed no additive phenotype (Fig 5A and 5B). This reinforces that the interactions of PAM2L motifs of Upa1 are the major determinants for endosomal attachment of Rrm4.

Next, we investigated endosomal shuttling. In strains expressing Rrm4-M1,2Δ-Kat missing MLLE1Rrm4 and MLLE2Rrm4 endosomal shuttling was not disturbed (Fig 5C). The number of motile Rrm4-M1,2Δ-Kat positive signals, their velocity, and their processivity were not affected (S8C and S8D Fig). Like above, we challenged the endosomal attachment of Rrm4 by expressing Upa1 versions with mutations in the PAM2L motifs. As expected, simultaneous mutation of both PAM2L motifs of Upa1 resulted in a reduction in the number of Rrm4-Kat positive shuttling endosomes (Figs 5C and S8E) [16]. When both PAM2L motifs were mutated, the Rrm4-Kat version was mislocalised and exhibited aberrant staining of filamentous structures in about 80% of hyphae (Fig 5C and 5D). This staining pattern was reminiscent of the microtubule association of Rrm4 previously observed during altered accumulation of static Rrm4-Kat in upa1Δ strains (S8C Fig) [43]. Quantifying Rrm4-Kat signals exhibiting processive movement in kymographs revealed that strains exhibiting aberrant staining of filamentous structures resulted in reduced fluorescence (Fig 5E) indicating fewer Rrm4-Kat versions on shuttling endosomes. As an important control, we treated the strains with the microtubule inhibitor benomyl, demonstrating that aberrant staining was microtubule-dependent (S9A Fig). Furthermore, Western blot analysis demonstrated that mutations in Rrm4 do not alter the protein amount (S9B Fig). Comparable to previous reports, we observed residual motility of Rrm4-Kat on shuttling endosomes if both PAM2L motifs were mutated or if upa1 was deleted (Fig 5C). This indicates additional proteins besides Upa1 are involved in the endosomal attachment of Rrm4 [16].

To analyse the influence of individual PAM2L motifs, we determined the number of hyphae with aberrant microtubule staining in strains co-expressing Rrm4-Kat versions and an Upa1-Gfp version with mutations of PAM2L sequence 1 or 2. Mutations in PAM2L1 and PAM2L2 caused 8% and 19% of hyphae with aberrant MT staining, respectively (Fig 5D). Hence, the interaction of PAM2L2 is more important for correct endosomal attachment of Rrm4. This is consistent with our biochemical results demonstrating that MLLE3Rrm4 binds stronger to PAM2L2 of Upa1 than to PAM2L1 (Fig 3D).

Next, we investigated the association of Rrm4-M1,2Δ-Kat in strains expressing Upa1 with mutated PAM2L1. In this strain, the endosomal attachment was solely dependent on the interaction of MLLE3Rrm4 with the PAM2L2 sequence of Upa1. We did observe 6% of hyphae with aberrant MT staining (Figs 5D; S7D). This was comparable to strains expressing Rrm4-Kat, suggesting no clear difference (Fig 5D). However, testing Upa1 with its PAM2L2 mutated, leaving only PAM2L1 for interaction with Rrm4, we observed a clear increase in hyphae with aberrant MT staining when comparing strains co-expressing Rrm4-M1,2Δ-Kat versus Rrm4-Kat (52% versus 19%; Fig 5D and 5E). Hence, the region covering MLLE1Rrm4 and MLLE2Rrm4 was important for Rrm4 attachment. Finally, we tested individual deletions in

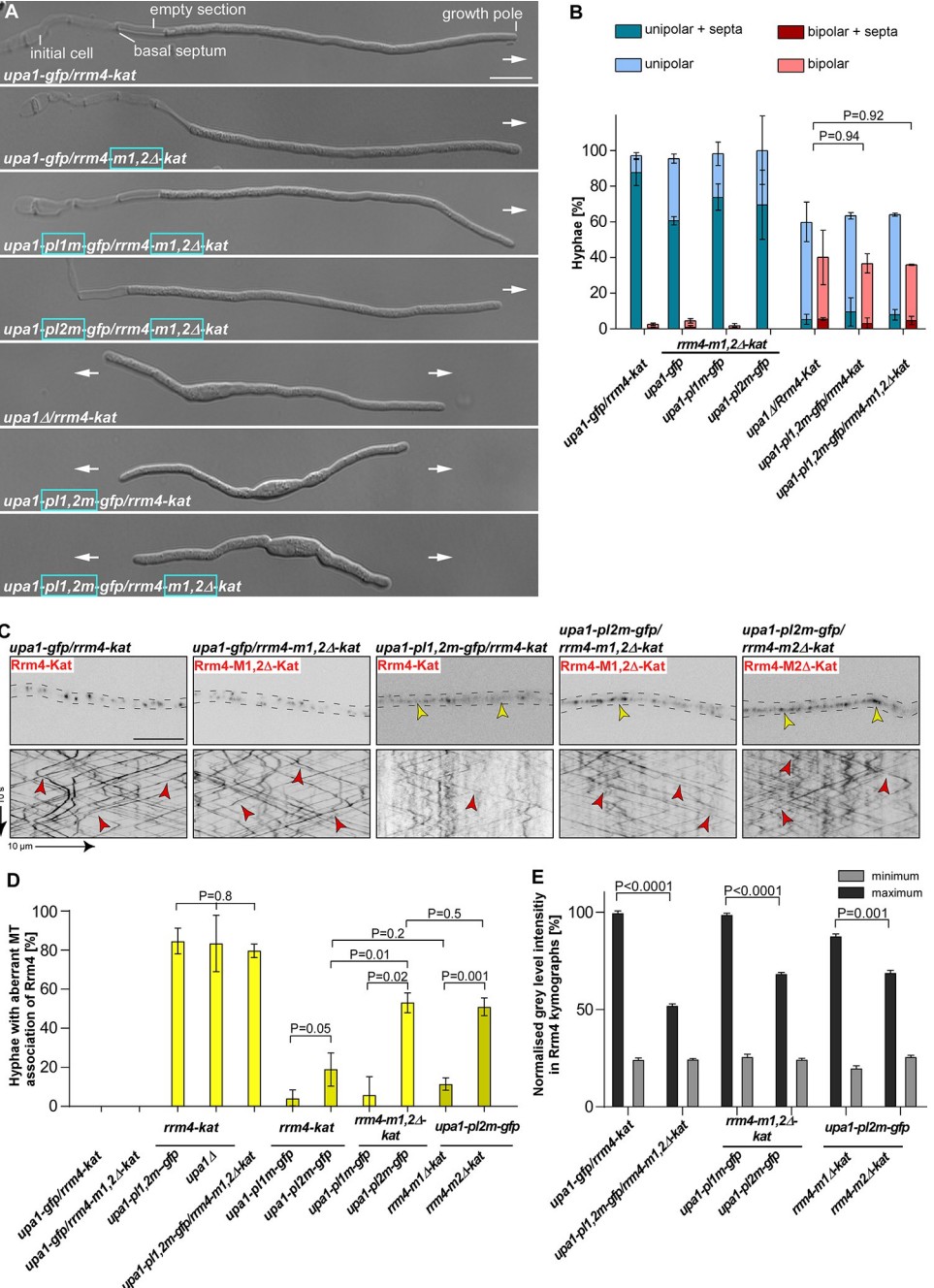

**Fig 5. MLLE2 plays an accessory role in endosomal attachment of Rrm4.** (**A**) Growth of AB33 derivatives in their hyphal form (6 h.p.i.; size bar 10 μm). Growth direction is marked by arrows. (**B**). Quantification of hyphal growth of AB33 derivatives shown in A (6 h.p.i.): unipolarity, bipolarity and basal septum formation were quantified (error bars, SEM; n = 3 independent experiments, > 100 hyphae were analysed per strain; For statistical analysis, the percentage of uni- and bipolarity was investigated by using unpaired two-tailed Student's t-test (α<0.05). (**C**) Micrographs (inverted fluorescence image; size bar, 10 μm) and corresponding kymographs of AB33 hyphae derivatives (6 h.p.i.) co-expressing various Upa1-Gfp and Rrm4-Kat versions as indicated. Movement of Rrm4-Kat versions is shown (arrow length on the left and bottom indicates time and distance, respectively). Bidirectional movement is visible as diagonal lines (red arrowheads). Aberrant microtubule staining is indicated by a yellow arrowhead. (**D**) Percentage of hyphae (6 h.p.i.) exhibiting aberrant microtubule association as indicated in panel C and S8C Fig. Set of strains that were analysed simultaneously are shown in the same yellow shading (error bars, SEM; for statistical evaluation, the percentage of hyphae with abnormal microtubule staining was compared by using unpaired two-tailed Student's t-test (α<0.05); n = 3 independent experiments, > 25 hyphae were analysed per strain). (**E**) Normalised minimum and maximum grey level intensities of shuttling signals measured in Rrm4 kymographs showed in Figs 5C and S5C (error bars, SEM; n = 3 independent experiments, 100 shuttling signals kymographs were analysed per strain, two-tailed Student's t-test (α > 0.05).

MLLE1[Rrm4] and MLLE2[Rrm4] in combination with mutated PAM2L2 in Upa1 to dissect the role of the different MLLE domains. In strains expressing Rrm4-Kat or Rrm4-M1Δ-Kat with this type of Upa1 mutation, the number of hyphae with aberrant MT staining was comparable (18% versus 11%, respectively; Fig 5D). However, strains expressing Rrm4-M2Δ-Kat exhibited an increased number of hyphae with aberrant MT staining that was comparable to Rrm4-M1,2Δ-kat (51% versus 52% respectively; Figs 5D; S7B). As mentioned above, aberrant MT localisation of mutated Rrm4-M2Δ-Kat and Rrm4-M1,2Δ-Kat also exhibited reduced intensity of processive signals in Rrm4 kymographs (Fig 5E) suggesting that the endosomal association was altered. To conclude, for MLLE1[Rrm4], we were unable to assign a clear function yet. However, MLLE2[Rrm4] plays an accessory role in the endosomal attachment of Rrm4. In essence, the C-terminus of Rrm4 contains three MLLE domains, with MLLE2[Rrm4] fulfilling an accessory role and MLLE3[Rrm4] having an essential function during the attachment of mRNPs to endosomes.

## Discussion

Combining structural biology and biophysical techniques with fungal genetics and cell biology, we addressed how mRNPs can be mechanistically linked to endosomes in the model fungus *U. maydis*. Previously, it was found that the C-terminal MLLE domain of Rrm4 is needed for shuttling [15] and that the C-terminus of Rrm4 interacts with two PAM2L motifs of Upa1 [16]. Now, we demonstrate that this region of Rrm4 contains not only two MLLE domains, but a sophisticated binding platform consisting of three MLLE domains with MLLE2 and MLLE3 functioning in linking the key RNA transporter to endosomes. We disclose a strict hierarchy with main and accessory domains. The accessory MLLE2 domain shows variations in the critical region of the predicted PAM2 binding pocket, suggesting a novel mode of interaction with currently unknown interaction partners. Rrm4 represents the first protein containing multiple MLLE domains to form a binding platform to the best of our knowledge. This interaction unit is essential for the correct endosomal attachment and, hence, mRNP trafficking.

### The MLLE / PAM2 connection

The founding member of the MLLE domain family is present at the C-terminus of the poly (A)-binding protein PABPC1. This domain interacts with PAM2 motifs of numerous interaction partners such as GW182, eRF3, and the RNA-binding protein LARP4 functioning in microRNA biology, translational termination, and posttranscriptional control, respectively [23,29,44]. Structural analysis revealed a common mode of binding, where the Leu and particularly the Phe of the PAM2 consensus motif xxLNxxAxEFxP (S3A Fig) are interacting with helix 2 and 3 as well as helix 3 and 5 of MLLE domain, respectively [28,29]. Indeed, the interaction of MLLE with a hydrophobic amino acid is highly conserved, which in most cases is Phe with a known exception in the variant PAM2w motif of LARP4 and LARP4A, where Trp is found (S3A Fig) [28,44,45].

Studying the MLLE domain-containing protein Rrm4, we discover that it has three MLLE domains in its C-terminal half. MLLE3[Rrm4] binds PAM2L motifs of Upa1 with a $K_D$ of 5 and 15 μM for PAM2L2[Upa1] and PAM2L1[Upa1], respectively. The binding affinities are in the same range as described for other MLLE/PAM2 interactions: for example, the binding of MLLE-[PABPC1] with PAM2[LARP1], PAM2[Tob2−125], PAM2[LARP4] exhibit a $K_D$ of 3.8, 16 and 22 μM, respectively [41]. Importantly, our biophysical assessment confirms the exquisite binding specificity of MLLE[Rrm4] that recognises PAM2L1[Upa1] and PAM2L2[Upa1] but not the PAM2[Upa1] version. PAM2L sequences contain a stretch of acidic amino acids in the N-terminal half, and the spacing of FxxP in the core sequence is altered (S3A Fig). These variations might account for the

differential binding mode. Visual inspection of the potential PAM2L binding region in the predicted model revealed that MLLE3$^{Rrm4}$ contains a Gly at position 736 to sustain the binding of an aromatic residue of PAM2L as described for other MLLE domains (see above). However, we were unable to uncover the structural basis for the sequence specificity. Towards this end, future structural studies are required to provide detailed information on the interaction of MLLE3$^{Rrm4}$ with PAM2L sequences.

Differential PAM2 binding has also been described for the MLLE$^{UBR5}$. This MLLE domain interacts with PAM2$^{PAIP}$ with an affinity of 3.4 μM [46], whereas it binds a PAM2L sequence (S3A Fig) in its own HECT domain with lower affinity ($K_D$ of 50 μM). The latter interaction has been implicated in regulating the HECT ligase activity [36]. Interestingly, the PAM2L sequence within the HECT domain of UBR5 is highly similar to the PAM2L1 and −2 of Rrm4: (i) the sequences contain an acidic stretch N-terminal to the conserved Phe (S3A Fig), (ii) the distance between Phe and Pro is two instead of one amino acid, and (iii) the PAM2L sequence contains an additional bulky Tyr in close vicinity to the Phe residue. Remarkably, MLLE$^{PABC1}$ does not recognise the PAM2L sequence of UBR5 [36]. In essence, although the strong sequence specificity of MLLE$^{Rrm4}$ and MLLE$^{Pab1}$ from *U. maydis* is, to the best of our knowledge, so far unique, we hypothesise that differential PAM2 and PAM2L interactions are evolutionarily conserved and might be more widespread than currently anticipated.

We also observed a clear binding specificity for MLLE$^{Pab1}$ from *U. maydis* that interacts with PAM2$^{Upa1}$ but not the PAM2L sequences from Upa1 (Fig 6). MLLE$^{Pab1}$ binds with comparable affinity to the PAM2$^{Upa1}$ ($K_D$ of about 14 μM, S3E Fig). Previously, we showed that mutations in PAM2$^{Upa1}$ strongly decreased MLLE$^{Pab1}$ binding but did not interfere with the endosomal shuttling of Pab1 [16]. Thus, there might be other members of the endosomal mRNPs interacting with Pab1 and stabilising its endosomal association. In fact, the dimerising scaffold protein Upa2 of endosomal mRNA transport contains four PAM2 motifs offering eight potential PAM2 motifs for interaction with Pab1 (Fig 6). However, mutating all four PAM2 motifs did not interfere with endosomal mRNA transport, although interaction with MLLE$^{Pab1}$ was lost [43], confirming a potential redundancy. Consistently, mutations in PAM2 of human LARP4B did not interfere with the function of stress granule recruitment, suggesting additional factors in this case [45].

Studying the other two MLLE domains of Rrm4 revealed that both lack the canonical Gly for interactions with PAM2 or PAM2L sequences. MLLE1$^{Rrm4}$ and MLLE2$^{Rrm4}$ have Ser471 and Arg573 instead, respectively. Consistently, MLLE1$^{Rrm4}$ and MLLE2$^{Rrm4}$ do not bind PAM2 or PAM2L sequences. Thus, although the general fold of the MLLE domain is probably conserved in MLLE1$^{Rrm4}$ and MLLE2$^{Rrm4}$, these domains most likely exhibit a different binding specificity to their potential protein partner. Our detailed *in vivo* analysis revealed that MLLE2 carries out an accessory function for the correct attachment of Rrm4 during endosomal shuttling. In the case of MLLE1, we did not identify a clear function so far. However, we believe that all three MLLE domains are functionally important. This is supported by the fact that the presence of an MLLE binding platform with three MLLE domains is evolutionarily conserved. Even Rrm4 versions of the distantly related fungus *Rhizophagus irregularis* contains three MLLE domains (Mucoromycota, determined by AlphaFold) [31].

Studying the spatial arrangement of the three MLLE domains revealed that they form a highly flexible binding platform pertinent for the regulation of Rrm4 mRNP transport. This would allow for the simultaneous interaction of several binding partners and potential rearrangements like an induced fit after binding. Such a scenario might be crucial during the loading and unloading of mRNPs to endosomes. Noteworthy, the N-terminal RNA-binding domain consisting of three RRMs is clearly separated from the MLLE domains for endosomal attachment. This is comparable with the arrangement of RRM and MLLE domains in human

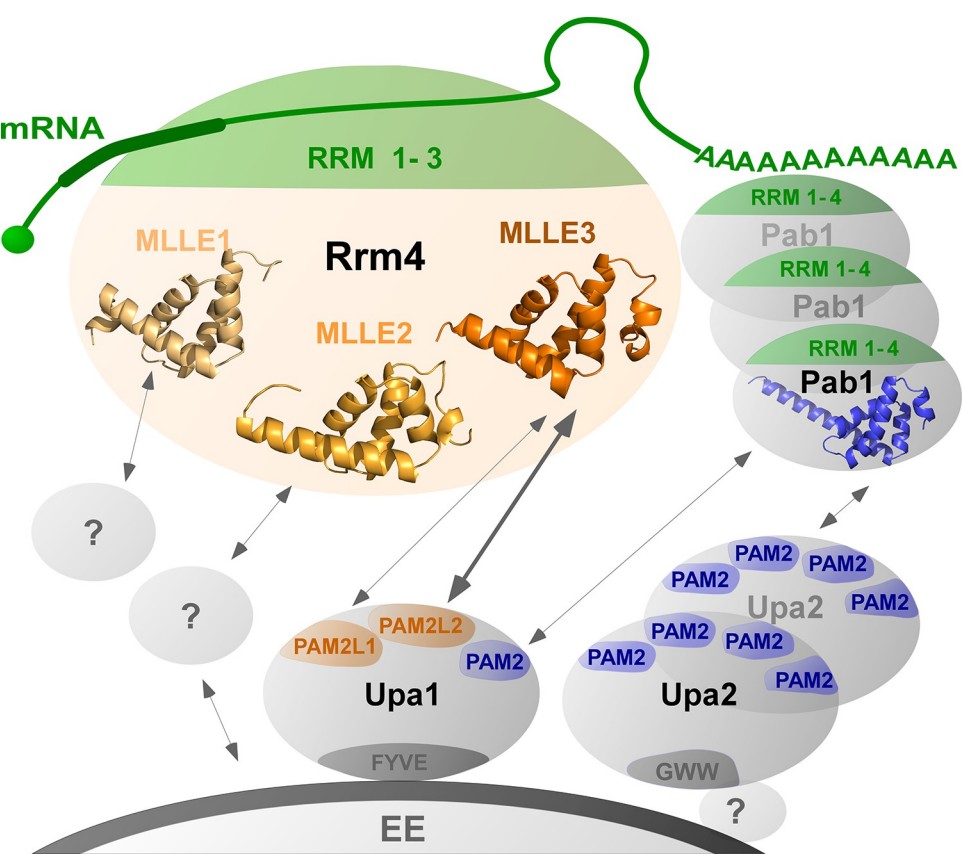

**Fig 6. Schematic model of endosomal attachment of mRNPs via MLLE domains.** Cargo mRNAs (green) are bound by the N-terminal RRM (1–3) domains of Rrm4 (green). The C terminal MLLE domains (orange) form a binding platform: MLLE3$^{Rrm4}$ interacts with PAM2L1$^{Upa1}$ and PAM2L2$^{Upa1}$(orange), MLLE1 and −2$^{Rrm4}$ might interact with currently unknown factors to support the endosomal binding. In particular, MLLE2$^{Rrm4}$ has an accessory role during endosomal interaction. The four RRMs of Pab1 (green) interact with the poly(A) tail, and the MLLE$^{Pab1}$ (blue) interacts with PAM2 of Upa1 and with the four PAM2 motifs of Upa2 (dark blue), a dimerising scaffold protein. Upa1 is attached to endosomes via its FYVE domain, and the C-terminal GWW motif of Upa2 is crucial for its endosomal binding.

PABPC1: the four N-terminal RRM domains interact with the poly(A) tail of mRNAs, and a flexible spacer region exposes the MLLE domain for protein/protein interactions [47]. Within the spacer region, additional interactions with the RRM2 of PABPC1 were found, suggesting a function in multimerization of the protein on the poly(A) tail of mRNAs [48].

## Conclusion

Endosomal mRNA transport is evolutionarily highly conserved. Besides hyphal growth in fungi, it is important for endosperm development in plants as well as neuronal functions in animals and humans [3,5,7,14]. Malfunctioning of this process causes defects in polar growth in fungi and has been implicated in neuronal diseases such as Charcot-Marie-Tooth type 2B neuropathy or amyotrophic lateral sclerosis in humans [8,9].

A key question is how mRNPs are linked to endosomes. In plants, two RRM-type RNA-binding proteins form a complex with cargo mRNAs and the endosomal component *N*-ethyl-maleimide-sensitive factor NSF as well as Rab5a [7,12,49]. Comparably, the FERRY complex (Five-subunit Endosomal Rab5 and RNA/ribosome intermediarY) interacts with the activated

GTP-bound form of Rab5 during endosomal mRNA transport in neurons [10,11]. Further examples are the membrane-associated protein Annexin 11 that links large RNA granules to lysosomal vesicles during mRNA transport in neuronal axons and dendrites [9]. Thus, a number of components and interactions are known, however detailed structural insights are scarce. Here, we have demonstrated that in hyphae, endosomal attachment of Rrm4 is mediated by an MLLE binding platform with a non-canonical accessory domain joining an essential MLLE domain for perfect interaction with Upa1 on the endosomal surface (Fig 6). This binary interaction in the core of the transport mRNPs is supported by numerous interactions of additional protein partners such as Upa2 and Pab1 that assist in attaching components to the endosomal surfaces (Fig 6). In closing, studying endosomal mRNP transport in fungal model systems might guide future research endeavours in plant and neuronal systems.

## Materials and methods

### Structural modelling of C-terminal MLLE domains of Rrm4

To obtain structural models of the C-terminal region of Rrm4, an iterative homology modelling approach was used with the TopModel workflow [32]. Initially, the entire C-terminal region (421 to 792) was submitted as input in TopModel and identified templates for MLLE3$^{Rrm4}$ (665–791 AA; Figs 1C, S1A). Then, the rest of the C-terminal part comprising amino acids 421 to 664 was resubmitted as input identifying other templates as a new starting point for the MLLE2$^{Rrm4}$ (571–629). Likewise, the remaining C-terminal sequence comprising amino acids 421 to 549 was resubmitted as input, identifying other templates as a new starting point for the MLLE1$^{Rrm4}$ (446–530). In total, this led to the identification of three MLLE domains, for which structural models were generated using default TopModel parameters. The quality of the structural models was assessed with TopScore [33].

### Plasmids, strains, and growth conditions

For molecular cloning of plasmids, *Escherichia coli* Top10 cells (Thermofisher C404010) and for recombinant protein expression *E.coli Lobstr* cells (Kerafast EC1002) were used respectively. Sequence encoding H-Rrm4-NT4 was inserted into the pET22 vector (Merck 69744) with an N-terminal hexa-histidine tag for crystallisation studies. Sequence encoding MLLE variants were inserted into the pGEX-2T vector (Merck GE28-9546-53) containing GST sequence in N-terminus for pulldown and ITC experiments. Sequence encoding PAM2 variants were inserted into the Champion pET-Sumo vector (Thermofisher K30001). pRarepLys plasmid was co-transformed in *E. coli Lobstr strain* to supplement the rare codons for efficient recombinant protein production. *E. coli* transformation, cultivation, and plasmid isolation were conducted using standard techniques. For yeast two-hybrid analyses *S. cerevisiae* strain AH109 (Matchmaker 3 system, Clontech) was used. Yeast cells were transformed and cultivated using standard techniques. All *U. maydis* strains are derivatives of AB33, in which hyphal growth can be induced by switching the nitrogen source in the medium [42]. *U. maydis* yeast cells were incubated in complete medium (CM) supplemented with 1% glucose, whereas hyphal growth was induced by changing to nitrate minimal medium (NM) supplemented with 1% glucose, both at 28˚C [42]. Detailed growth conditions and general cloning strategies for *U. maydis* are described elsewhere [6,50,51]. All plasmids were verified by sequencing. Strains were generated by transforming progenitor strains with linearised plasmids. Successful integration of constructs was verified by diagnostic PCR and by Southern blot analysis [50]. For ectopic integration, plasmids were linearised with *Ssp*I and targeted to the *ip*$^S$ locus [52]. A detailed description of all plasmids, strains, and oligonucleotides is given in S3–S9 Tables. Sequences are available upon request.

## Recombinant protein expression and purification

*E. coli* cells from freshly transformed plates were inoculated in 20 ml expression media. To produce high-density expression cultures with tight regulation of induction and expression in shake flasks we designed a complex media inspired by the principle of Studier's autoinduction media [53]. We use adequate amount of glucose to prevent the unintended induction and leaky expression of target protein as well as phosphate buffer to prevent acidity as a result of glucose metabolism from the excessive glucose in the media. In addition, the medium contained glycerol, nitrogen, sulphur, and magnesium for promoting high-density growth. Unlike the Studier's autoinduction media our media lack lactose therefore expression can be induced with IPTG and expressed at required temperature (1.6% Trypton, 1% Yeast extract, 50 mM $Na_2HPO_4$, 50 mM $KH_2PO_4$, 25 mM $[NH_4]_2SO_4$, 0.5% Glycerol, 0.5% Glucose, 2 mM $MgSO_4$) [53] with ampicillin (100 mg/ml) and chloramphenicol (34 mg/ml) or kanamycin (200 mg/ml) and chloramphenicol (34 mg/ml) and grown overnight (16 hours) at 37˚C, 200 rpm. Note that the high concentration of kanamycin was used to prevent the unintended resistance promoted by high phosphate concentration [53]. Supernatant from the overnight culture was removed by centrifugation at 4˚C, 5000 × g for 2 minutes. Cells were resuspended in fresh media with a starting $OD_{600}$ of 0.1 and grown at 37˚C, 200 rpm for about 2 hours 30 minutes until the $OD_{600}$ = 1. Protein expression was induced at 28˚C, 200 rpm, for 4 hours by addition of 1 mM IPTG, and harvested by centrifugation at 4˚C, 6,000 × g for 5 minutes. Protein purification was performed as per the previous report [54]. Hexa-histidine tagged protein was purified using Nickel-based affinity chromatography (HisTrap HP, GE Healthcare) on Akta prime FPLC system. Cells were thawed on ice and resuspended in buffer A (20 mM HEPES pH 8.0, 200 mM NaCl, 1 mM EDTA, 10 mM Imidazole pH 8.0; 1 mM PMSF, 0.5 mg/ml Lysozyme, 0.5 mg/ml DNase, 1mM β mercaptoethanol [β-ME]). Subsequently, cells were lysed by sonication on ice and centrifuged at 4˚C 18,000 × g for 30 minutes. Resulting supernatant was loaded onto a pre-equilibrated column with buffer B (20 mM HEPES pH 8.0, 200 mM NaCl,10 mM Imidazole), washed with buffer C (20 mM HEPES pH 8.0, 200 mM NaCl, 50 mM Imidazole, 1 mM β-ME), eluted with buffer D (20 mM HEPES pH 8.0, 200 mM NaCl, 300 mM Imidazole, 1mM β-ME) and further purified by size exclusion chromatography (HiLoad 26/600 Superdex 200, GE Healthcare), pre-equilibrated with buffer E (20 mM HEPES pH 8.0, 200 mM NaCl, 1 mM β-ME). For crystallisation studies, H-Rrm4-NT4 was purified as above except that the buffers were prepared with high salt (500 mM NaCl) and without β-ME.

GST-tagged protein was purified using Glutathione-based affinity chromatography (GSTrap FF GE Healthcare). Cells were thawed on ice and resuspended in Buffer F (20 mM HEPES pH 8.0, 200 mM NaCl, 1 mM EDTA, pH 8.0; 1 mM PMSF, 0.1 mg/ml Lysozyme, 1 mM β-ME). Subsequently, cells were lysed by sonication on ice and centrifuged at 4˚C, 18,000 g for 30 minutes. The resulting supernatant was loaded onto a pre-equilibrated column with buffer E (20 mM HEPES pH 8.0, 200 mM NaCl, 1 mM β-ME) and washed with the same buffer, eluted with buffer H (20 mM HEPES pH 8.0, 200 mM NaCl, 10 mM reduced glutathione, 1 mM β-ME), and further purified by size exclusion chromatography (HiLoad 16/600 Superdex 200 GE Healthcare), pre-equilibrated with buffer E. Protein purity was analysed on SDS-PAGE. All the purified proteins were concentrated, centrifuged at 4˚C, 100,000 × g for 30 minutes, quantified by Nanodrop A280, aliquoted, and stored at −80˚C. Peptides for ITC experiments were custom-synthesised and purchased from Genscript, USA (see Fig 3A for peptide sequence).

## GST pull-down experiments

Pull-down assays were performed as per the previous report [43]. In short, GST-MLLE variants and HS-PAM2 variants were expressed in *E. coli*. Cell pellets from 50 ml *E. coli* expression

culture were resuspended in 10 ml buffer F (20 mM HEPES pH 8.0, 200 mM NaCl, 1 mM EDTA; 0.5% Nonidet P-40, 1 mM PMSF, 0.1 mg/ml Lysozyme). Cells were lysed by sonication on ice and centrifuged at 4˚C, 16,000 × g for 10 minutes. 1 mL of the resulting supernatant was incubated for 1 hour at 4˚C on constant agitation of 1,000 rpm with 100 μL glutathione sepharose (GS) resin (GE Healthcare), pre-equilibrated in buffer F. The GS resin was washed three times with 1 ml of buffer G (20 mM HEPES pH 8.0, 200 mM NaCl, 1 mM EDTA, 0.5% Nonidet P-40). Subsequently, supernatant of HS-PAM2 variants was added to the GST-MLLE variant bound resins and incubated for 1 hour at 4˚C on agitation. The resins were washed as aforementioned, resuspended in 100 μL of 2x Laemmli loading buffer, boiled for 10 minutes at 95˚C and analysed by western blotting.

## *Ustilago maydis* cell disruption and sample preparation for immunoblotting

*U. maydis* hyphae were induced as described earlier (see Plasmids, strains, and growth conditions). 50 ml of hyphal cells (6 h.p.i) were harvested in 50 ml conical centrifuge tubes by centrifugation at 7,150 × g, for 5 minutes. Cell pellets were resuspended in 2 ml phosphate-buffered saline pH 7.0 (PBS; 137 mM NaCl, 2.7mM KCl, 8 mM $Na_2HPO_4$ and 2 mM $KH_2PO_4$) and transferred to a 2 ml centrifuge tubes. Cells were harvested at 7,150 × g for 5 minutes and supernatant was removed completely. The resulting cell pellets were flash-frozen in liquid nitrogen and stored at −80˚C until use. Sample tubes were placed on 24 well TissueLyser adapter (Qiagen 69982) and soaked in liquid nitrogen for 1 minute, 5 mm stainless steel bead was added to each sample tube and the cells were disrupted at 30 Hz for 3 times 1 minute in Mixer Mill MM400 (Retsch, Germany), with intermittent cooling between shaking. At the end of the cell disruption dry homogenised powder of cells was resuspended in 1 ml urea buffer (8 M urea, 50 mM Tris/HCl pH 8.0 containing one tablet of 'cOmplete' protease inhibitor per 25 ml, Roche, Germany; 1 mM DTT; 0.1 M PMSF) and centrifuged at 16,000 × g for 10 minutes at 4˚C. The supernatant was used for subsequent analysis. Samples were diluted ten times and protein concentrations were measured by BCA assay (Thermofisher 23225). Samples were diluted to 1 mg/ml final concentration in Laemmli buffer and boiled at 95˚C for 10 minutes. 40 μg of each sample was loaded in 1.5 mm thickness gels for SDS-PAGE, subsequently analysed by Western blotting.

## Yeast two-hybrid analysis

Yeast two-hybrid analyses were performed as per the previous report (16). The two-hybrid system Matchmaker 3 from Clontech was used as per manufacturer's instructions. Yeast strain AH109 was co-transformed with derivatives of pGBKT7-DS and pGADT7-Sfi (S8 Table, S4 Fig) and were grown on synthetic dropout plates (SD) without leucine and tryptophan at 28˚C for 2 days. Transformants were patched on SD plates without leucine and tryptophan (control) or on SD plates without leucine, tryptophan, histidine, and adenine (selection). Plates were incubated at 28˚C for 2 days to test for growth under selection conditions. For qualitative plate assays, cells (SD -leu, -trp, OD600 of 0.5) were serially diluted 1:5 with sterile water, spotted 4 μl each on control and selection plates and incubated at 28˚C for 2 days. Colony growth was documented with a LAS 4000 imaging system (GE Healthcare).

## SDS-PAGE and immunoblotting

All SDS–PAGE and Western blotting experiments were performed as reported previously [43]. Western blotting samples were resolved by 8 or 10 or 12% SDS-PAGE and transferred

and immobilised on nitrocellulose membrane (Amersham Protran) by semi-dry blotting using Towbin buffer (25 mM Tris pH 8.6, 192 mM Glycine, 15% Methanol).

Proteins were detected using α -His from mouse (Sigma H1029), α-Gfp from mouse, (Roche, Germany), α-tRfp from rabbit (AB233-EV, Evrogen) and α-Actin from mouse (MP Biomedicals, Germany) as primary antibodies. As secondary antibodies α-mouse IgG HRP conjugate (Promega W4021) or α-rabbit IgG HRP conjugate (Cell Signaling #7074) were used. Antibodies bound to nitrocellulose membranes were removed by incubating in TBS buffer pH 3.0 (50 Tris pH 3.0, 150mM NaCl) at room temperature, before detecting with the constitutively expressed control (α-Actin). Detection was carried out by using ECL$^{TM}$ Prime (Cytiva RPN2236). Images were taken by luminescence image analyser, LAS4000 (GE Healthcare) as per the manufacturer's instructions.

### Multiangle light scattering (MALS)

MALS was performed as per the previous report [55]. Superdex 200 Increase 10/300 GL column (GE Healthcare) was pre-equilibrated overnight at 0.1 ml/minute flow rate with buffer E (20 mM HEPES pH 8.0, 200 mM NaCl, 1 mM β-ME). For each analysis, 200 μl of a protein sample at 2.0 mg/ml concentration was loaded onto the column at 0.6 ml/minute flow rate using a 1260 binary pump (Agilent Technologies). The scattered light was measured with a miniDAWN TREOS II light scatterer, (Wyatt Technologies), and the refractive index was measured with an Optilab T-rEX refractometer, (Wyatt Technologies). Data analysis was performed with ASTRA 7.3.2.21 (Wyatt Technologies) [56].

### Crystallisation of H-Rrm4 NT4

Initial crystallisation conditions were searched using MRC 3 96-well sitting drop plates and various commercially available crystallisation screens at 12˚C. 0.1 μl homogeneous protein solution (10 mg/ml in 20 mM Hepes pH 8.0, 500 mM NaCl) was mixed with 0.1 μl reservoir solution and equilibrated against 40 μl of the reservoir. After one week, initial rod-shaped crystals were found, which were then further optimised by slightly varying the precipitant concentrations. Optimisation was also performed in sitting drop plates (24-well) at 12˚C but by mixing 1 μl protein solution with 1 μl of the reservoir solution, equilibrated against 300 μl reservoir solution. Best diffracting crystals were grown within 7 days in 0.1 M Hepes pH 7.5, 20% (w/v) PEG 10000 (Qiagen PEG I, D5). Before harvesting the crystal, crystal-containing drops were overlaid with 2 μl mineral oil and immediately flash-frozen in liquid nitrogen.

### Data collection, processing, and structure refinement

A complete data set of the H-Rrm4-NT4 were collected at beamline ID23EH1 (ESRF, France) at 100 K and wavelength 0.98 Å up to 2.6 Å resolution. All data were processed using the automated pipeline at the EMBL HAMBURG and reprocessed afterwards using XDS [57]. Above obtained model for MLLE2 by TopModel was successfully used to phase the 2.6 Å data set of Rrm4 MLLE using the program Phaser from the program suite Phenix [58]. The structure was then refined in iterative cycles of manual building and refinement in Coot [59], followed by software-based refinements using the program suite Phenix [58]. All residues were in the preferred and additionally allowed regions of the Ramachandran plot (S2 Table). The data collection and refinement statistics are listed in S2 Table. The structure and models were compared using the superpose tool of PHENIX to calculate the corresponding RMSD. The images of the models were prepared using PyMOL. The structure was deposited at the worldwide protein data bank under the accession code 7PZE.

## Small-angle X-ray scattering

We collected all SAXS data on beamline BM29 at the ESRF Grenoble [60]. The beamline was equipped with a PILATUS 2M detector (Dectris) with a fixed sample to a distance of 2.827 m. To prevent concentration-dependent oligomerisation, we performed the measurements with 0.6 mg/ml protein concentrations at 10˚C in buffer E. We collected one frame each second and scaled the data to absolute intensity against water. All used programs for data processing were part of the ATSAS Software package (Version 3.0.3) [61]. The primary data reduction was performed with the program Primus [39]. With Primus and the included Guinier approximation [62], we determined the forward scattering $I(0)$ and the radius of gyration ($R_g$). The pair-distribution function $p(r)$ was calculated with Gnom [63] and was used to estimate the maximum particle dimension ($D_{max}$). Due to the high flexibility of the proteins we performed an Ensemble Optimization Method (EOM) [38]; default parameters, 10,000 models in the initial ensemble, native-like models, constant subtraction allowed) with the predicted MLLE domains from TopModel [32,64] for H-Rrm4-NT4 and G-Rrm4 with an additional GST (PDB1ua5). We uploaded the data to the Small Angle Scattering Biological Data Bank (SASBDB) [65,66] with the accession codes SASDMS5(G-Rrm4) and SASDMT5 (H-Rrm4-NT4).

## Isothermal titration calorimetry

All ITC experiments were performed as per the previous report [54]. All the protein samples used in ITC were centrifuged at 451,000 × g for 30 minutes and quantified by Nanodrop (A280) before use. The concentration of GST or His-tagged MLLE variants was adjusted to 30 μM and PAM2 peptide variants was adjusted to 300 μM using buffer G (20 mM HEPES pH 8.0, 200 mM NaCl, 1 mM 2 ME). Using an MicroCal iTC200 titration calorimeter (Malvern Panalytical technologies), a PAM2 peptide variant with a volume of 40 μL was titrated to the different GST-MLLE variants. All experiments were repeated at least twice. ITC measurements were performed at 25˚C with 40 injections (1 μL each). Only the first injection had a volume of 0.5 μL and was discarded from the isotherm. The other technical parameters were reference power, 5 μcal s$^{-1}$; stirring speed, 1000 rpm; spacing time, 120 s, and a filter period, 5 s. The resulting isotherm was fitted with a one-site binding model using MicroCal Origin for ITC software (MicroCal LLC). Note, that the binding of GST-Rrm4-NT4 and H-Rrm4-NT4 were comparable indicating that tagging of the Rrm4 versions did not influence the binding affinity (Figs 3B; S5D).

## Microscopy, image processing and image analysis

Laser-based epifluorescence-microscopy was performed on a Zeiss Axio Observer.Z1 as previously described [43]. Co-localisation studies of dynamic processes were carried out with a two-channel imager (DV2, Photometrics, Tucson, AZ, USA) [67]. To quantify uni- and bipolar hyphal growth, cells were grown in 30 ml cultures to an $OD_{600}$ of 0.5, and hyphal growth was induced. After 6 hours, more than 100 hyphae were analysed per strain towards their growth behaviour (n = 3). Cells were assessed for unipolar and bipolar growth as well as the formation of a basal septum. To analyse the signal number, velocity, and travelled distance of fluorescently labelled proteins, movies with an exposure time of 150 ms and 150 frames were recorded. More than 25 hyphae were analysed per strain (n = 3). To inhibit microtubule polymerisation, hyphal cultures were incubated with 50 μM of benomyl (Sigma Aldrich) for 2 h at 28˚C and 200 rpm [15]. All movies and images were processed and analysed using the Metamorph software (Version 7.7.0.0, Molecular Devices, Seattle, IL, USA). For the generation of kymographs, 20 μm of hyphal cell starting at the hyphal tip were used. To determine the

minimum and maximum grey level intensities of shuttling endosomes, 100 signals were analysed per strain (the ten most prominent signals per kymograph that showed processive movement of > 20 μm without changes in directions were chosen per strain). The minimum and maximum grey level intensities were measured using the region measurement tool of the Metamorph software. All pixel intensities were measured, and minimum as well as maximum intensities for each region were listed (16-bit images). The grey level intensities were normalised to the wild-type intensity, which was set to 100%. For statistical analysis of the signal number, velocity, and travelled distance, processive signals with a travelled distance of more than 5 μm were conducted and counted manually. For determination of aberrant microtubule staining, hyphae were counted manually as well. Data points represent means from three independent experiments ($n$ = 3) with mean of means (red line) and SEM. For all statistical evaluations, two-tailed Student´s $t$-tests were used. Determination of strains exhibiting aberrant staining of microtubules was scored manually. For verification, key comparisons were evaluated independently by two experimentalists. Importantly, the key findings were confirmed (S5H Fig). We used the data obtained by the more experienced microscopist in the main figure (Fig 5D). All evaluated data are compiled in Spreadsheat S1.

## Supporting information

**S1 Fig. The presence of three MLLEs is verified by additional modelling predictions.** (**A**) Compilation of MLLE sequences used for modelling with the highest similarity of MLLE1-3$^{Rrm4}$. (**B**) Structural models obtained with TopModel overlaid to Rrm4 full-length models obtained with the recently available tools as indicated. Natural alignments between corresponding MLLE domains have an RMSD < 2Å, mutually confirming the quality of the independently modelled structures. The differences in the relative domain arrangements in both full-length models and the disordered regions in between the domains suggest a high mobility within Rrm4.
(TIF)

**S2 Fig. The three MLLEs of Rrm4 are located in a flexible C-terminal region.** (**A**) Schematic representation of protein variants drawn to scale (molecular weight in kilo Dalton indicated next to protein bar) using the following coloring: dark green, RNA recognition motif (RRM); orange, MLLE$^{Rrm4}$ domains; (**B**) SDS PAGE analysis of purified G-Rrm4, H-Rrm4-NT4 used in crystallography and SAXS measurement. (**C**) MALS-SEC analysis of H-Rrm4-NT4. Graph shows the elution profile. Dotted line in red indicate the apparent molecular weight as observed in the light scattering. (**D**) $R_g$ distribution calculated by EOM pool is shown in grey bars and the selected models in blue bars *left* GST_Rrm4 *right* H-Rrm4NT4 (**E**) *Left* Selected models of the EOM analysis for GST-Rrm4. The MLLE subdomains and the GST are shown in cartoon representation (MLLE1 in light orange, MLLE2 in orange, MLLE 3 in dark orange) and the missing amino acids as grey spheres. **I:** The model has a $R_g$ of 8.94 nm, a $D_{max}$ of 29.22 nm with a volume fraction of~0.38. **II:** The model has a $R_g$ of 8.75 nm, a $D_{max}$ of 23.99 with a volume fraction of~0.25. **III:** The model has a $R_g$ of 7.74 nm, a $D_{max}$ of 25.90 with a volume fraction of~0.12. **IV:** The model has a $R_g$ of 8.33 nm, a $D_{max}$ of 28.79 with a volume fraction of~0.12. **V:** The model has a $R_g$ of 9.14 nm, a $D_{max}$ of 33.73 with a volume fraction of~0.12. *Right* Selected models of the EOM analysis for H-Rrm4NT4. The MLLE subdomains are shown in cartoon representation (MLLE1 in light orange, MLLE2 in orange, MLLE3 in dark orange) and the missing amino acids as grey spheres. **I:** The model has a $R_g$ of 5.12 nm, a $D_{max}$ of 15.56 with a volume fraction of~0.17. **II:** The model has a $R_g$ of 5.90 nm, a $D_{max}$ of 18.73 nm with a volume fraction of~0.08. **III:** The model has a $R_g$ of 5.10 nm, a $D_{max}$ of 16.43 nm with a

volume fraction of~0.75.
(TIF)

**S3 Fig. MLLE1 $^{Rrm4}$, $-2^{Rrm4}$ are not essential for PAM2L1 $^{Upa1}$ and -L2 $^{Upa1}$ binding in GST pull-down assay.** (**A**) Comparison of PAM2 sequences found in Upa1 (UniprotKB ID A0A0D1E015) with those of human proteins, such as Usp10 (Q14694), GW182 Q9HCJ0), Mkrn1 (Q9UHC7), Paip1 (Q9H074), Paip2 (Q9BPZ3), Atx2 (-Q99700), NFX (Q12986), eRF3 (P15170), PAN3 (Q58A45), LARP4 (Q71RC2), LARP4b (Q92615), Tob (P50616), HECT (O95071), Asp and Glu are indicated in red stressing the highly negative charges in PAM2L sequences. (**B**) Western blot analysis of GST co-purification experiments with components expressed in *E. coli*: N-terminal Hexa-Histidine-SUMO-tagged PAM2 variants were pulled down by N-terminal GST fused MLLE variants of Rrm4 and Pab1. Experiment was performed with the soluble fraction of *E. coli* cell lysate to demonstrate specific binding. Results were analysed with αHis immunoblotting.
(TIF)

**S4 Fig. MLLE1 $^{Rrm4}$, $-2^{Rrm4}$ are not essential for PAM2L1 $^{Upa1}$ and -L2 $^{Upa1}$ binding in yeast two-hybrid experiments.** Yeast two-hybrid analyses with schematic representation of protein variants tested on the left. Cultures were serially diluted 1:5 (decreasing colony-forming units, cfu) and spotted on respective selection plates controlling transformation and assaying reporter gene expression (see Materials and methods).
(TIF)

**S5 Fig. MLLE $^{Pab1}$ does not bind PAM2L1 $^{Upa1}$ and -L2 $^{Upa1}$.** (**A**) SDS-PAGE analysis of purified GST-MLLE variants used in ITC experiments (see also S6 Fig). (**B**) Representative isothermal titration calorimetry (ITC) binding curves of MLLE $^{Pab1}$ domain. Experiments were performed using GST or hexa-histidine-tagged MLLE variants and synthetic PAM2 and PAM2L peptide variants. $K_D$ values of two independent measurements are given (values corresponding to the indicated data are given in bold).
(TIF)

**S6 Fig. MLLE1 $^{Rrm4}$, $-2^{Rrm4}$ do not contribute to the binding of PAM2L1 $^{Upa1}$ and -L2 $^{Upa1}$.** (**A-C**) Representative isothermal titration calorimetry (ITC) binding curves of MLLE domains. Experiments were performed using GST or hexa-histidine-tagged MLLE variants and synthetic PAM2 and PAM2L peptide variants. $K_D$ values of two independent measurements are given (values corresponding to the indicated data are given in bold). (**D**) Summary of ITC results shown in Figs 3 and S3. $K_D$ values are given in μM.
(TIF)

**S7 Fig. Deletion of MLLE3 $^{Rrm4}$ abolishes endosomal movement of Rrm4.** (**A-C**) Quantification of processive Rrm4-Kat (top) and Upa1-Gfp signals (bottom; (**A**)), velocity of fluorescent Rrm4-Kat (top) and Upa1-Gfp signals (bottom; (**B**)) and the travelled distance of processive Rrm4-Kat (top) and Upa1-Gfp signals (bottom; (**C**); per 10 μm of hyphal length; only particles with a processive movement of $>$ 5 μm were conducted; data points representing mean from n = 3 independent experiments, with mean of means, red line and SEM; unpaired two-tailed Student's t-test (α$<$0.05), for each experiment at least 25 hyphae were analysed per strain). (**D**) Summary of the *in vivo* analysis is shown in Figs 4,5 and S7–S10.
(TIF)

**S8 Fig. Deletion of MLLE1 $^{Rrm4}$ and $-2$ cause aberrant staining of microtubules.** (**A**) Growth of AB33 derivatives in their hyphal form (6 h.p.i.; size bar 10 μm). Growth direction is marked by arrows. (**B**) Quantification of hyphal growth of AB33 derivatives shown in panel A (6 h.p.

i.): unipolarity, bipolarity and basal septum formation were quantified (error bars, SEM.; n = 3 independent experiments, > 100 hyphae were analysed per strain; For statistical evaluation, the percentage of uni- and bipolarity was investigated by using unpaired two-tailed Student's t-test (α<0.05). (**C**) Micrograph and Kymograph of AB33 hyphae derivates (6 h.p.i.) expressing red and green fluorescent proteins as indicated. Fluorescence signals were detected simultaneously using dual-view technology (arrow length on the left and bottom indicates time and distance, respectively). Processive co-localising signals are marked by red arrowheads. Aberrant microtubule staining is indicated by a yellow arrowhead. (**D-E**) Quantification of processive Rrm4-Kat signals (left), velocity of fluorescent Rrm4-Kat (middle) and the travelled distance of processive Rrm4-Kat signals (right) related to Fig 5C and EV5C, respectively (per 10 μm of hyphal length; only particles with a processive movement of > 5 μm were conducted; data points representing mean from n = 3 independent experiments, with mean of means, red line and SEM; unpaired two-tailed Student's t-test (α<0.05), for each experiment at least 25 hyphae were analysed per strain).
(TIF)

**S9 Fig. Mislocalisation of Rrm4 is microtubule-dependent.** (**A**) Benomyl treatment is shown in micrograph and kymograph of AB33 hyphae derivates (6 h.p.i.) expressing red and green fluorescent proteins. Processive signals, as well as static signals, post benomyl treatment are marked by red arrowheads. Aberrant microtubule staining is indicated by a yellow arrowhead. (**B**) Western blot analysis of the expression levels of Rrm4 and Upa1 variants 6 h.p.i. of hyphal growth. Rrm4 and Upa1 variants were detected via mKate2 and Gfp, respectively. Actin was detected as loading control. Bands representing full-length proteins are marked with arrows.
(TIF)

**S10 Fig. Independent testing of aberrant microtubule staining.** Evaluation of the most important strains showing aberrant microtubule staining analysed by two experimentalists (we used the data obtained by the more experienced microscopist in Fig 5D; see Materials and methods).
(TIF)

**S1 Table. Accession numbers for protein sequences used in multiple sequence alignment of MLLE domains.**
(RTF)

**S2 Table. Data collection and refinement statistics.**
(RTF)

**S3 Table. SAXS data.**
(RTF)

**S4 Table. Description of *U. maydis* strains used in this study.**
(RTF)

**S5 Table. Generation of *U. maydis* strains used in this study.**
(RTF)

**S6 Table. Description of plasmids used for *U. maydis* strain generation.**
(RTF)

**S7 Table. Description of plasmids used for recombinant expression in *E. coli*.**
(RTF)

**S8 Table. Description of plasmids used for yeast two-hybrid analyses.**
(RTF)

**S9 Table. DNA oligonucleotides used in this study.**
(RTF)

**S1 Spreadsheet. Microscopic data used for quantifications in Figs 4, 5, S7, S8 and S10.**
(XLSX)

## Acknowledgments

We thank Drs. Florian Altegoer and Dierk Niessing as well as laboratory members for critical reading of the manuscript. We acknowledge Drs. Georg Groth, Alexander Minges, and Mohanraj Gopalswamy for support with ITC experiments. We gratefully acknowledge the computational support provided by the "Center for Information and Media Technology" (ZIM) at the Heinrich Heine University Düsseldorf and the computing time provided by the John von Neumann Institute for Computing (NIC) on the supercomputer JUWELS at Jülich Supercomputing Centre (JSC) (user IDs: HKF7, VSK33). The synchrotron MX data were collected at ESRF ID23-EH1. We thank Sylvain Engilberge for the assistance in using the beamline. The final SAXS experiments were performed on beamline BM29 at the European Synchrotron Radiation Facility (ESRF), Grenoble, France. We are grateful to Dihia Moussaoui at the ESRF for assisting with data collection.

## Author Contributions

**Conceptualization:** Senthil-Kumar Devan, Stephan Schott-Verdugo, Lilli Bismar, Lutz Schmitt, Sander HJ Smits, Holger Gohlke, Michael Feldbrügge.

**Data curation:** Senthil-Kumar Devan, Stephan Schott-Verdugo, Kira Müntjes, Lilli Bismar, Jens Reiners, Astrid Höppner, Sander HJ Smits, Michael Feldbrügge.

**Funding acquisition:** Sander HJ Smits, Holger Gohlke, Michael Feldbrügge.

**Investigation:** Senthil-Kumar Devan, Stephan Schott-Verdugo, Kira Müntjes, Eymen Hachani.

**Methodology:** Kira Müntjes, Lilli Bismar, Jens Reiners, Eymen Hachani, Astrid Höppner.

**Project administration:** Sander HJ Smits, Holger Gohlke, Michael Feldbrügge.

**Software:** Jens Reiners.

**Supervision:** Kira Müntjes, Lilli Bismar, Lutz Schmitt, Sander HJ Smits, Holger Gohlke, Michael Feldbrügge.

**Validation:** Lutz Schmitt, Astrid Höppner, Sander HJ Smits, Holger Gohlke, Michael Feldbrügge.

**Writing – original draft:** Senthil-Kumar Devan, Stephan Schott-Verdugo, Sander HJ Smits, Holger Gohlke, Michael Feldbrügge.

**Writing – review & editing:** Sander HJ Smits, Holger Gohlke, Michael Feldbrügge.

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
