## [Decision Letter · Decision Letter 0]

6 May 2022

Dear Dr Feldbrugge,

Thank you very much for submitting your Research Article entitled 'A MademoiseLLE domain binding platform links the key RNA transporter to endosomes' to PLOS Genetics.

The manuscript was fully evaluated at the editorial level and by independent peer reviewers. The reviewers appreciated the attention to an important topic but identified some concerns that we ask you address in a revised manuscript

We therefore ask you to modify the manuscript according to the review recommendations. Your revisions should address the specific points made by each reviewer.

[LINK]

Yours sincerely,

Aaron P. Mitchell, PhD

Associate Editor

PLOS Genetics

Gregory P. Copenhaver

Editor-in-Chief

PLOS Genetics

Reviewer's Responses to Questions

**Comments to the Authors:**

Reviewer #1: The authors have satisfactorily addressed all my comments from the Review Commons evaluation process. I also believe they have done a good job of addressing the comments of the other reviewers. I now recommend publication provided the following very minor points are addressed:

1. On line 295 the authors conclude that the pattern in microtubule-related but only show the evidence (benomyl treatment) that this is the case later in this section. The authors should fix this, for example by first pointing out that the pattern was reminiscent of the microtubule association seen in the previous study and then confirming that the signal was derived from microtubules with benomyl.

2. Mislocalised is spelled incorrectly on line 294.

3. There are some inconsistencies with the tense that need fixing, e.g.:

Line 199. "However, when both PAM2L1,2 motifs were mutated, the interaction between the Upa1 and Rrm4 is lost….."

There are other examples of conflicting tenses that need resolving.

Reviewer #2: The manuscript by Devan et al. has already been reviewed by three individuals through the Review Commons process. All three reviewers considered that the work represents a significant advance, as it provides important new information about MLLE and PAM interactions that is relevant to the broader field of endosome-mediated RNA trafficking. I concur with this opinion. The previous three reviewers agreed that experimental evidence for the role of MLLE2 in facilitating Rrm4 association with endoscopes needed to be strengthened. I consider that the new experimental data included in Figures 5 and EV5 satisfactorily address this concern. I also think that the various minor revisions requested by the previous reviews have been fully addressed.

Reviewer #3: Devan, Scott-Verdugo et al. performed a structure-function study characterizing the interaction between Rrm4 and Upa1, two proteins required for mRNA transport on early endosomes in Ustilago maydis. The authors use structural prediction, structural biology, and biochemistry to determine that Rrm4 contains three MLLE motifs, with MLLE3 being required for association with two PAM2-like motifs within Upa1. They then perform cell biology in Ustilago maydis and confirm that MLLE3 motif is required for Rrm4 association with motile early endosomes. Finally, they find that the MLLE2 motif of Rrm4 also plays a role in tethering Rrm4 to early endosomes. These findings demonstrate an adaptable binding platform between Rrm4 and Upa1, suggesting potential complex regulation of RNA transport on early endosomes.

This submission was received by PLOS Genetics as a Review Commons article and has already undergone revision following review from three reviewers (prior to this review). I read the submission and previous revisions. This is an interesting, well-written, thorough study. The previous reviews were adequately addressed and the article should be accepted as-is.

**Have all data underlying the figures and results presented in the manuscript been provided?**

Reviewer #1: Yes

Reviewer #2: Yes

Reviewer #3: Yes

PLOS authors have the option to publish the peer review history of their article (what does this mean?). If published, this will include your full peer review and any attached files.

Reviewer #1: No

Reviewer #2: **Yes: **PAUL LASKO

Reviewer #3: No

---

## [Editor Report · Decision Letter 1]

20 May 2022

Dear Dr Feldbrugge,

We are pleased to inform you that your manuscript entitled "A MademoiseLLE domain binding platform links the key RNA transporter to endosomes" has been editorially accepted for publication in PLOS Genetics. Congratulations!

Yours sincerely,

Aaron P. Mitchell, PhD

Associate Editor

PLOS Genetics

Gregory P. Copenhaver

Editor-in-Chief

PLOS Genetics

Comments from the reviewers (if applicable):

**Data Deposition**

http://datadryad.org/submit?journalID=pgenetics&manu=PGENETICS-D-22-00463R1

**Press Queries**

---

## [Editor Report · Acceptance letter]

15 Jun 2022

PGENETICS-D-22-00463R1 

A MademoiseLLE domain binding platform links the key RNA transporter to endosomes 

Dear Dr Feldbrügge, 

We are pleased to inform you that your manuscript entitled "A MademoiseLLE domain binding platform links the key RNA transporter to endosomes" has been formally accepted for publication in PLOS Genetics! Your manuscript is now with our production department and you will be notified of the publication date in due course.

With kind regards,

Zsofia Freund

PLOS Genetics

On behalf of:
